# Improved pore structure characterization and classification of strong diagenesis sandstones by data-mining analytics in Tazhong area, Tarim Basin

**Feng Tian**[1,2], **Xidong Wang**[1,2]*, **Xinyi Yuan**[1,2], **Di Wang**[1,2]

**1** School of Geological and Mining Engineering, Xinjiang University, Urumqi, Xinjiang, China, **2** Xinjiang Key Laboratory for Geodynamic Processes and Metallogenic Prognosis of the Central Asian Orogenic Belt, Xinjiang, China

* microdifficult@xju.edu.cn

**Data Availability Statement:** All relevant data are within the manuscript and its Supporting information files.

## Abstract

The Silurian system in Tazhong area is characterized by extensive, low-abundance lithological reservoirs with strong diagenesis, resulting in significant heterogeneity. The complex pore structure in this area significantly impacts fluid control, making accurate characterization and classification of pore structures crucial for understanding reservoir properties and their influence on oil and gas distribution. Based on 314 Mercury Injection Capillary Pressure (MICP) samples in combination with core slices and thin casting slices observation, a pipeline of characterization and classification scheme by data-mining analytics of strong diagenesis sandstone pore structure types in the study zone is established, and the characteristics of different pore structures are clarified. According to the pore structure parameter abstracted by MICP data compression and variable analysis based on hierarchical clustering and principal component analysis (PCA) analysis, the variables are reasonably evaluated and screened, and the screened variables can be divided into three groups: mean pore throat radius-maximum pore throat radius-median pore throat radius-pore throat diameter mean variable group, microscopic mean coefficient variable group, and median pressure displacement pressure-relative sorting coefficient variable group. The combination of classification schemes analysed by decision tree model and linear discriminant analysis (LDA) model was determined. In the two-dimensional projection diagram of LDA model, a relatively obvious distribution of low displacement pressure, middle displacement pressure and high displacement pressure was obtained, and three distribution lines were nearly parallel. Based on the relevant information, 6 combined classification schemes suitable for final pore structure modelling were determined verified by microscopic observation. The correct characterization and classification of pore structure can be applied to the prediction of pore type, which can be used to improve the prediction of oil and gas distribution and oil and gas recovery in the future.

**Funding:** The study is funded by Tianchi talent project (Grant No.40300-23005104). The funders had a role in study design and decision to publish mainly.

**Competing interests:** The authors have declared that no competing interests exist.

## 1. Introduction

The pore structure in tight sandstone is pivotal, directly impacting fluid flow, permeability, and hydrocarbon recovery efficiency, thus playing a crucial role in optimizing production strategies and maximizing resource recovery. The current research trend in the characterization and classification of pore structure in tight sandstone reservoirs involves the utilization of methods such as high-pressure mercury intrusion [1], high-resolution 3D X-ray micro-CT [2], electron microscopy, and advanced computational modeling [3]. Characterization of the pore structure and the method of classification scheme is always a challenge in low permeability reservoir evaluation. Despite numerous options, no unified methodology has reached consensus due to geological setting differences [4, 5]. Methods to characterize the pore structure include casting thin section observation combined with mercury injection and relative permeability data statistical analysis [6], using T2 spectrum of nuclear magnetic resonance (NMR) data fitting microscopic pore structure of pseudo capillary force curve, micro CT and multi-dimensional imaging technology, including digital core model based on formula Kozeny Carman [7, 8], among others. However, these methods come with various disadvantages [9]. The analysis of mercury injection data is based on statistical description and regression, which often does not reflect the real pore type; the application of nuclear magnetic technology can accurately characterize the pore type, microscopic CT and other multi-dimensional imaging technology [10] is the most direct and accurate method to reflect the pore type of reservoir [11], but both NMR and CT imaging technologies exhibit certain limitations. NMR technology faces challenges in accurately characterizing complex pore networks indirectly, processing intricate data, and restrictions on sample types. Meanwhile, CT technology suffers from constraints in sensitivity, potential radiation damage, and sample preparation. Both techniques are also associated with high equipment and operational costs, they cannot be widely used, the same limitations are mainly due to the fusion problem between the technology and large-scale tools [12, 13]. The pore structure of study zone in Tazhong area is affected by strong diagenesis, and different sand bodies pore space in tidal flat faces are transformed from primary pores to the secondary or remained pores with cementation and dissolution by undergoing different diagenesis stages [14, 15]. It is of great significance to characterize and classify the pore structure in data-based dimension and less subjectivity.

Therefore, we have applied a data-mining oriented characterization and classification pipeline [16–18] in fully utilization of an abundant 314 MICP samples dataset in the study zone with the tool of data mining mainly by LDA model and decision tree model. The pore structure was pre-classified into 15 types based on morphology method. The combined classification scheme was established by comparing the LDA model and the decision tree model, merging the original classification into 6 categories. We discovered the principle of the distribution trend on the projection diagram of the different classes and characterize the type in data dimension and geological dimension. In the two-dimensional projection diagram of the LDA model, a relatively clear distribution of low, middle, and high displacement pressures was observed, with the three distribution lines being nearly parallel. By combination with core slices observation, we characterize the pore types. It reflects the complexity of the strong diagenesis pore structure and microscopic heterogeneity in this area.

## 2. Geological setting

The study area is located in the central uplift belt of Tarim Basin as shown in Fig 1. with Manjiaer depression to the north and Central uplift to the south including the main Tazhong low uplift and Gugubi uplift at the western dip end of Tadong low uplift. The area of study is mainly around TZ11 and TZ12 wellblocks. The stratigraphic formation can be divided into

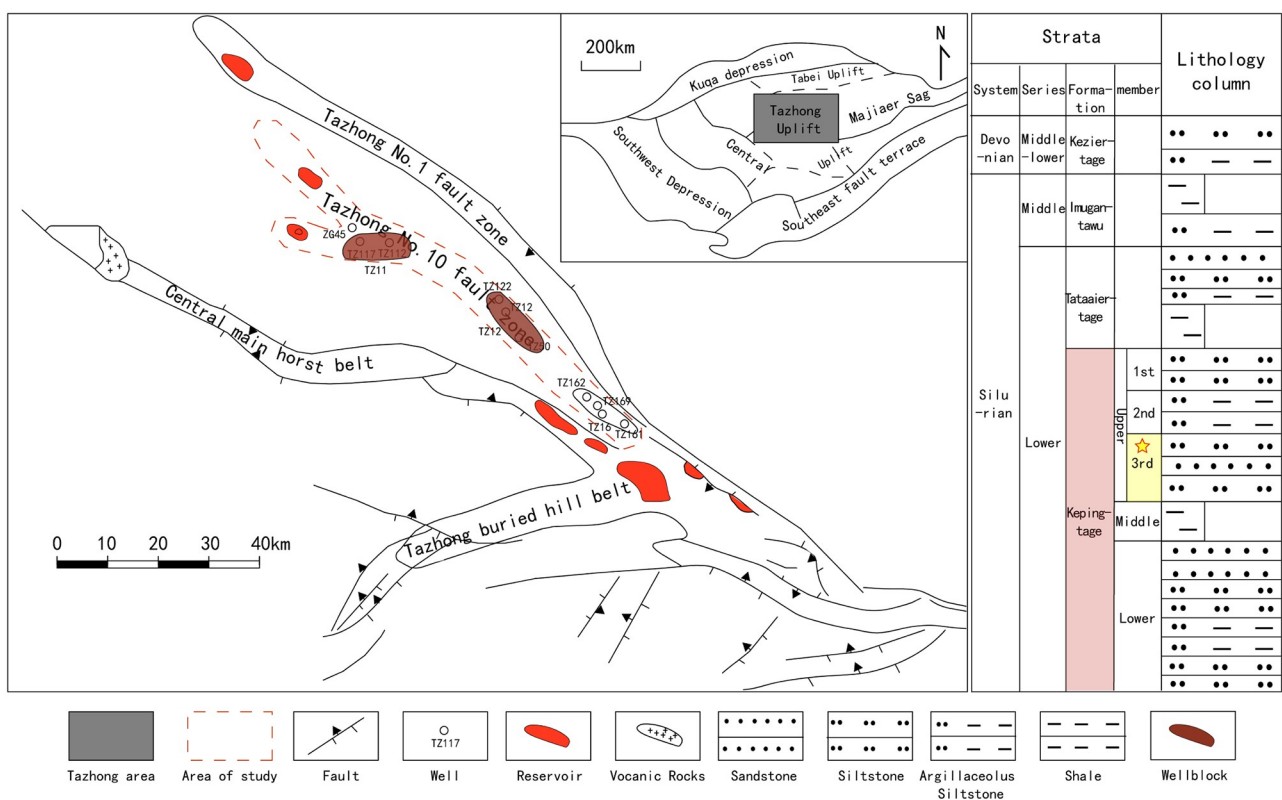

**Fig 1. Geological structures of Tazhong area and studied stratum.**

three groups and five lithological members from top to bottom [15], which are named as Imugantawu Formation, Tataertag Formation, upper Kepingtag, middle Kepingtag and lower Kepingtag member. In the Tazhong area, there are 52 exploration wells encountering relatively stable Silurian strata. The Imugantawu Formation consists of thick layers of brown mudstone interbedded with thin to thick layers of fine sandstone. The Tataertag Formation mainly comprises brown interbedded sandstone and mudstone, with some dominated by red mudstone. The upper Kepingtag member consists of gray-green thin to medium bedded fine sandstone, shale, and siltstone interbeds. The middle Kepingtag member is primarily gray-green medium bedded shale with interbeds of thin sandstone and silty sandstone. The lower Kepingtag member comprises gray-green medium bedded fine sandstone interbedded with silty sandstone, sandy shale, and thin shale layers. Upper-3rd submember of upper Kepingtag is the zone of study. Sedimentary facies in Tazhong area have been conducted implying the subtidal zone, intertidal zone (sand flat, mixed flat, mud flat) and supratidal zone deposits appeared successively from sea to land along the depositional tendency [14, 15, 19].

## 3. Methodology

### 3.1 Classification and characterization methods of pore structure

The MICP measurements are used to assess pore-throat structures [20]. The capillary pressure curves as well as parameters such as displacement pressure, maximum mercury intrusion saturation, medium pore-throat radius, and maximum pore throat radius ($R_{max}$) were obtained [21, 22]. The pore systems consist of pores connected through throats, and MICP analysis can

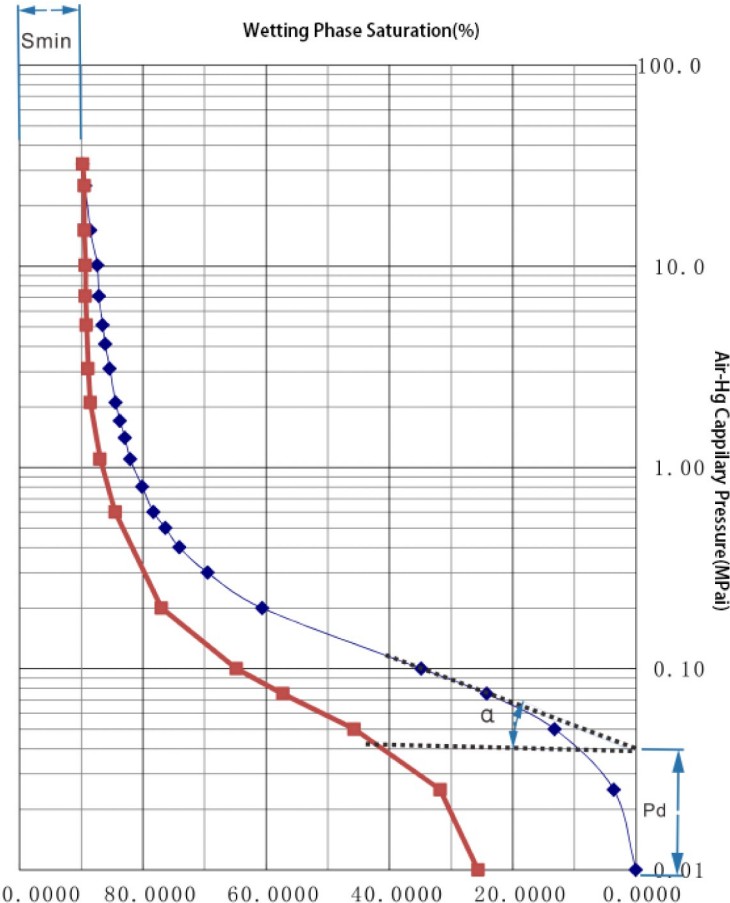

**Fig 2. Mercury injection curve and morphological classification parameters.**

provide insights into the pore structure and pore-throat size distributions [23–27]. The mercury injection capillary pressure (MICP) experimental data for the 314 samples utilized in this study were acquired through laboratory MICP experiments conducted on the samples upon their arrival at the laboratory.

By morphological classification, it was used to determine the pore structure type according to the displacement pressure $P_d$, minimum unsaturation volume percentage $S_{min}$ (usually described by the maximum mercury influx percentage, i.e., 1-$S_{min}$) and gentle to steep trend $\alpha$ angle reflected by the mercury injection curve as shown in Fig 2 [28, 29]. $P_d$ shows the capillary pressure corresponding to $R_d$, the largest radius connected pore throat in the pore system. $S_{min}$ is influenced by many factors related to the maximum pressure exerted by micropore volume. The $\alpha$ angle is mainly related to sample sorting.

The commonly used classification criteria are shown in Table 1 [30, 31], which can ensure the requirements of statistical description of morphological classification results and can be used for the understanding of statistical description of pore structure and quantitative research on classification.

### 3.2 Pore structure parameters with formula and its physical meaning

Different pore structure parameters can be characterized by using MICP data. Pore structure parameters can quantitatively characterize pore throat radius, sorting, uniformity of pore

**Table 1. Morphological classification parameter criteria of mercury injection curve.**

| MICP Parameter | Low | Medium | High |
|---|---|---|---|
| $P_d$ | <0.1MPa | 0.1-1Mpa | >1Mpa |
| $1-S_{min}$ | <10% | 10%-80% | >80% |
| α | 0–15˚ | 15–25˚ | >25˚ |

throat distribution, connectivity, the degree of detour, etc. to reflect the pore comprehensively [32, 33]. The pore structure parameters can be calculated using various distribution function model. Table 2 is the calculated pore structure parameters with formula and its physical meaning:

### 3.3 Workflow of pore structure characterization and classification

For the MICP characteristic parameters, it is of importance to analyze on data-dimension of characteristic parameters. The collinearity of the characteristic variables was removed, the parameter optimization and compression were carried out, and the secondary evaluation was carried out on the basis of the pre-classified 15 pore structure type's schemes. Finally, the final classification scheme of pore structure based on quantitative parameter data analysis of mercury injection can be determined. The workflow is illustrated in Fig 3.

## 4. Results and discussion

### 4.1 Petrology characteristics of sandstone

The submember of study has fine grain size as shown in Fig 4. It can be classified into five rock types by grain size, medium sandstone, fine sandstone, siltstone and irregular sandstone.

The diagram in Fig 5(a) and 5(b) [34] shows the sandstones is dominated by lithic, with a small amount of quartz lithic sandstone, and few other rock types in the upper-3rd submember of Kepingtag Formation.

However, previous study shows a very high content of skeleton quartz [14], which is not consistent with the high content of debris indicated by the thin section analysis of the rock. Cathode luminescence indicates the components with different colours [6] with quartz brown and blue-purple light, the quartz secondary edge does not emit light, quartzite debris emits brown light, feldspar emits blue light, rock debris emits brown light, the kaolinite complex group emits indigo light, calcite cement emits orange yellow light or orange red light, iron dolomite does not emit light, and iron dolomite emits orange red light. Some of the samples in the prospecting wells were observed which tells the rocks are identified as quartz sandstone in the year of 1995. After verification of multiple data, the lithic composition was mainly composed of quartzite and a small amount of phyllite and felfelite. The "quartz sandstone" in the upper-3rd submember should be classified as lithic quartz sandstone in Fig 5(a) and 5(b). Cathode luminescence is shown in Fig 5(c) and 5(d). The cuttings are mainly quartz with blue-purple light, accounting for a relatively high proportion of quartzite cuttings. The quartz-dominated rock skeleton has strong compressive resistance and stable properties, which provides relatively stable skeleton conditions for Kepingtag Formation.

### 4.2 Classification and characterization of pore structure

The pore structure was pre-classified according to the morphological classification criteria. 314 samples were classified into 15 pore structure types according to displacement pressure (Pd, P for abbreviation), degree of sorting α (A for abbreviation)and percentage of mercury

**Table 2. Calculated pore structure parameters with formula and its physical meaning.**

| Calculated parameter | Definition | Physical meaning | Calculated formula |
|---|---|---|---|
| $r_{max}$ | Pore throat radius maximum (μm): | The pore throat radius of the non-wetting phase beginning to enter the rock is the maximum pore throat radius which is an important parameter to indicate the rock permeability. | $r_{max} = \frac{0.7354}{P_d}$ |
| $r_{50}$ | Pore throat radius median (μm): | The corresponding pore throat radius When the non-wetting phase saturation is 50%, which can approximately represent the average pore throat radius of the sample. | $r_{50} = \frac{0.7354}{P_{50}}$ |
| $\bar{r}$ | Pore throat radius average (μm): | It is a parameter indicating the size of the average pore throat radius of a rock. | $\bar{r} = \frac{\sum (r_{i-1} + r_i)(s_i - s_{i-1})}{2\sum (s_i - s_{i-1})}$ |
| $\alpha$ | Homogeneous coefficient: | The homogeneity coefficient characterizes the deviation degree of each pore throat from the maximum pore throat radius in the pore media of the reservoir rock | $\alpha = \frac{\sum_{i=1}^{n} \frac{r_i}{r_{max}} \times \Delta S_i}{\sum_{i=1}^{n} \Delta S_i} = \frac{1}{r_{max} \times S_{max}} \int_0^{S_{max}} r_{(s)} \times dS$ |
| $F$ | Apparent Formation Factor: | It is the ratio of measured permeability of rock sample with calculated permeability, which reflects the maze of throat. | $F = \frac{K}{0.0000111333 \, \phi \int_0^{S_{max}} r_{(S)}^2 ds}$ |
| $s_p$ | Calculated Sorting coefficient | This is a measure of the standard deviation of the pore throat size in the sample, which directly reflects the concentration of the pore throat distribution. | $S_p = \sqrt{\frac{\sum (r_i - \bar{r})^2 \times \Delta S_i}{\sum \Delta S_i}}$ |
| $\phi_p$ | Structure coefficient: | It represents the difference between the real rock pore characteristics and the imaginary length of the cylindrical parallel bundle with different thickness. Its numerical value is an indicator of the various synthetic factors that affect the difference. | $\phi_p = \frac{\phi}{8K}(\bar{r})^2$ |
| $D_r$ | Coefficient of variation: | It can better reflect the pore throat size distribution uniformity degree of parameters. The smaller the numerical value, the more uniform the pore throats distribute. | $D_r = \frac{s_p}{\bar{r}} = \frac{1}{\bar{r}}\sqrt{\frac{\sum (r_i - \bar{r})^2 \times \Delta S_i}{\sum \Delta S_i}}$ |
| $1/(\phi_p \cdot D_r)$ | Characteristic structure coefficient: | It is the reciprocal product of the coefficient of variation, Dr and the structure coefficient φp, which reflects both the degree of pore throat sorting and the connection degree of the pore throat. The smaller the value, the worse the pore structure acts. | $1/(\phi_p \cdot D_r)$ |
| $S_{kp}$ | Bias (skewness): | A parameter that represents the symmetry of the pore throat size distribution, when $S_{KP} = 0$ is symmetric, and the $S_{KP} > 0$ is biased; $S_{KP} < 0$ is a negative bias. | $S_{kp} = \frac{S_p^{-3} \times \sum (r_i - \bar{r})^3 \times \Delta S_i}{\sum \Delta S_i}$ |
| $K_p$ | Kurtosis: | A parameter that represents the steep degree of the pore throat distribution frequency curve, when Kp = 1 is a normal distribution curve; Kp >1 is a high peak curve; Kp <1 is a slow or doublet curve. | $K_p = \frac{S_p^{-4} \times \sum (r_i - \bar{r})^4 \times \Delta S_i}{\sum \Delta S_i}$ |
| $K_{j\_max}$ | Peak permeability distribution (%): | Pore throat radius with the largest permeability contribution rate | $K_j = \frac{\int_{S_j}^{S_{j+1}} r_{(S)}^2 dS}{\int_0^{S_{max}} r_{(S)}^2 dS}$ |
| $K_{j\_max\_posi}$ | Peak Permeability Distribution position (%): | The percentage of permeability that can be provided by the pore throat radius with the largest rate of permeability contribution. | |

$P_d$—Displacement pressure. MPa;

P50—Capillary pressure when mercury saturation is 50%. MPa;

Si—Mercury saturation in i. %;

ri—The throat radius at i.μm

$\Delta$Si—The mercury saturation corresponding to ri %;

K:Permeability μm$^2$; φ—Porosity %;

r(s)—Pore throat radius in distribution function. μm;

ds—Mercury saturation corresponding to an interval.%;

Smax—Cumulative mercury saturation at the highest pressure. %;

Smin—Residual mercury saturation when the mercury is removed to the initial pressure. %;

S—Mercury saturation %;

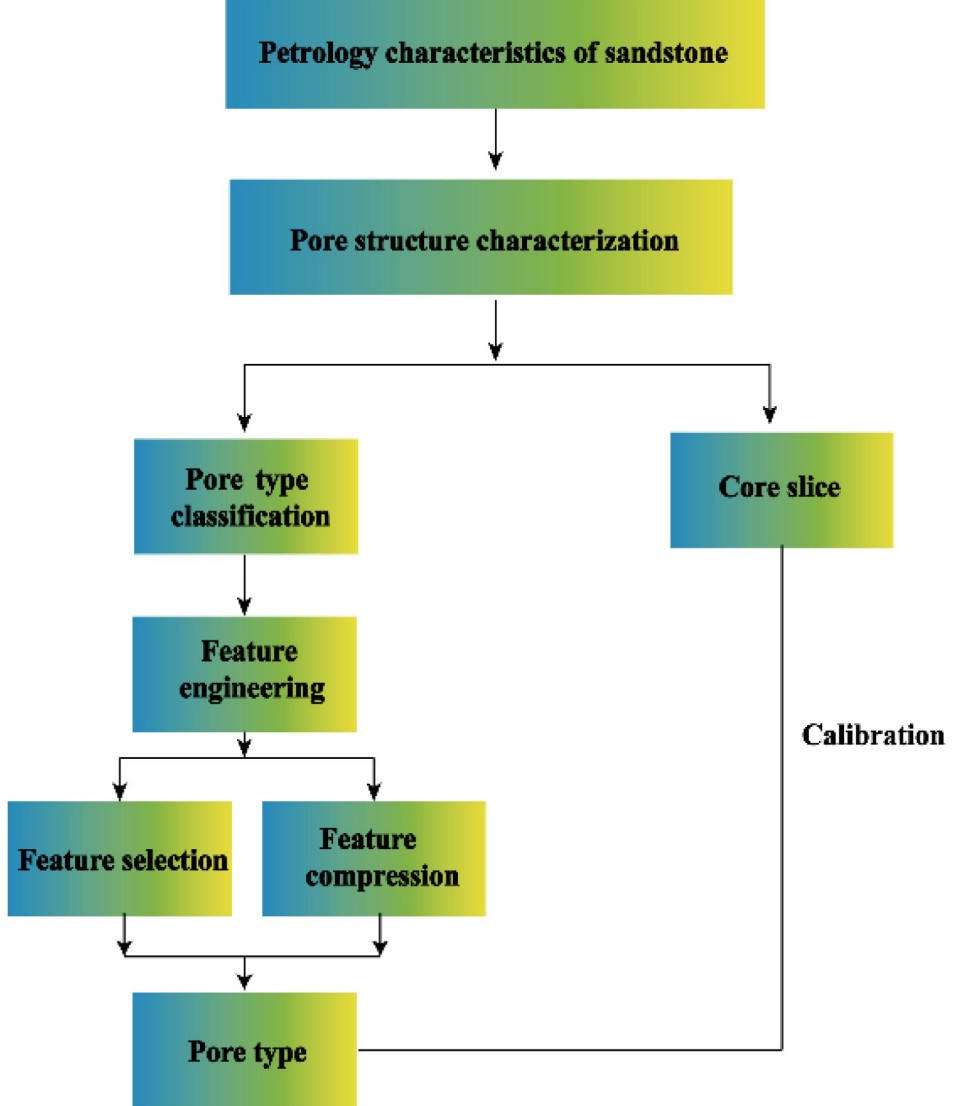

**Fig 3. Workflow of improved pore structure characterization and classification.**

injected volume (1-Smin) (S for abbreviation) as variables. We will use low (L)-middle (M)-high (H) for description the 3 parameters. The 15 types are low Pd-great α implying good sorting-high injected volume type which is used as Type LP-HA-HS, and the others are LP-MA-HS type, LP-MA-MS type, LP-LA-HS type, LP-LA-MS type, MP-HA-HS type, MP-MA-HS type, MP-MA-MS type, MP-LA-HS type, MP-LA-MS type, HP-MA-HS type, HP-MA-MS type, HP-LA-HS type, HP-LA-MS type, HP-LA-LS type. Fig 6 shows the statistics of different types of samples.

   The following rules can be consulted on the figure: 1) The categories should be 27 types, but the final sample shows 15, mainly because there are no more extreme pore structure types such as LP-LS or HP-HA.2) According to the statistics of different types of samples, LP-LA-HS and LP-HA-HS were the most developed. In combination with the geological conditions of this area, due to the influence of diagenesis, the cementation blockage and the development of

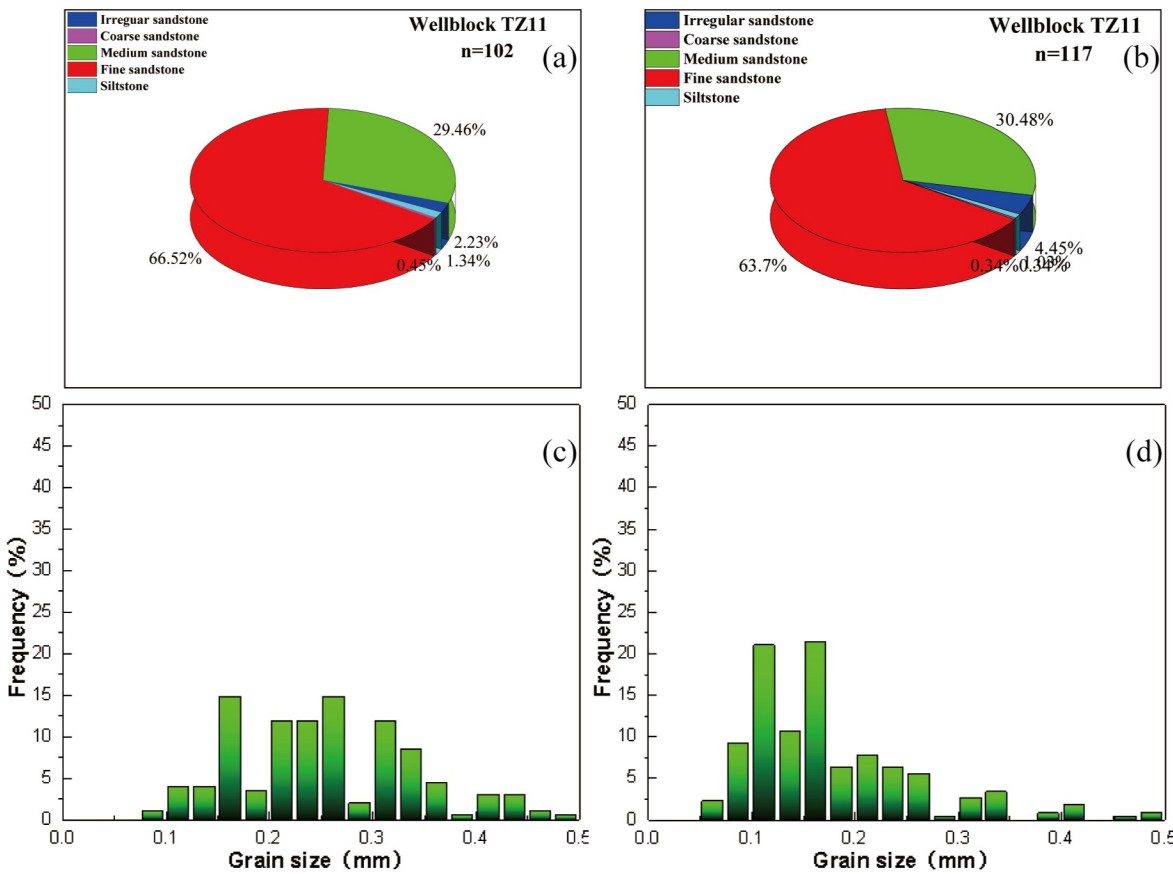

**Fig 4. Grain-size pie charts and bar plots of statistics of sandstone.**

dissolution pores will simultaneously produce the pore structure of large pore- thin throat pore throats type or large pore-coarse throat pore throats type. These pore structures are mainly residual primary pores and dissolution enlarged pores. This provides a geological data basis for the prevalence of these two types in statistics. 3) The transitional pore structure is also relatively developed. The pore types are mainly of MP -LA-HS type and HP-LA-MS. As a transition type with LP-LA-HS, MP-LA-HS type reflects the further reduction of the mean pore-throat radius. The HP-LA-MS type reflects the further transition to the worst structure type of HP-LA-LS type.4) The extreme type with less distribution is mainly the type of HP-LA-HS. High Pd and large mercury injection volume are contradictory, which reflects the presence of a small amount of more complex pore structure. At the same time, LP-LA-MS reflects the existence of large isolated pores, with more dead pores and less connected throats.

Corresponding typical MICP curves of pore structure, pore-throat distribution and corresponding pore-permeability histograms are shown in Fig 7. The following rules can be clearly obtained from the statistical chart: 1) According to the results of the low displacement pressure group Type 1–5, the main reason for the low displacement pressure was the existence of connected pore-throats >4μm, and the dominant pore-throats were more than 20% when the sorting was large, while the dominant pore-throats were not existed when the sorting was low. The proportion of pore-throats with different radius was average, and the highest was 10%. There were two kinds of pore throat with 10% to 20% ratio in the middle sorting type group, and the percentage of mercury injected volume was closely related to the porosity, and the

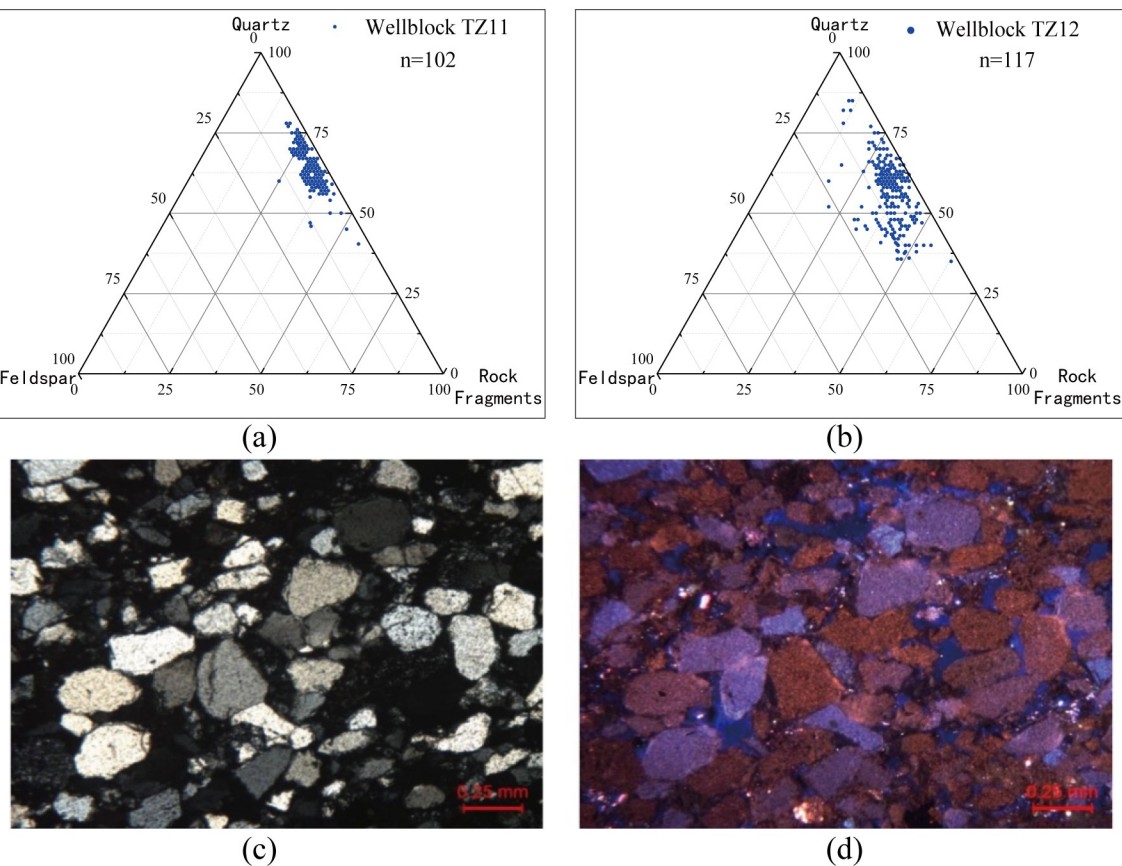

**Fig 5. Triangular diagram of sandstones components (a.b) and Cathode luminescence (CL) images(c.d).**

larger porosity was corresponded to maximum mercury injected volume. As the most dominant type of pore structure with low displacement, high sorting and high mercury injected volume, the pore permeability has certain characteristics: the porosity median is 14%, and the permeability is more than 70mD. The porosity of low displacement, low sorting and high mercury injection type is slightly smaller with a median distribution of 12%, and the permeability is more than 10mD. The pore permeability of other types of pore structures is not obvious, the porosity median is 10%, and the permeability is more than 1mD. 2) Results of the middle displacement pressure group Type 6–10; 1–2.5µm was the dominant developed grade pore throat radius. The sorting law is the same as the analysis of low Pd group, and the pore permeability law of each type of pore structure is not obvious, and the porosity median is mostly about 10%, and the difference is that the distribution is normal or skewed. Considering the problem of sample size, in the middle displacement pressure group, the high sorting and high mercury injection types showed an obvious normal distribution pattern, with the samples with 10% porosity being the most developed, while the middle mercury injection with low sorting showed an obvious left-skewed development. The samples with 10% porosity were also developed, and the samples with smaller porosity were also more distributed. The permeability showed a certain correlation with the volume of injected mercury volume. The permeability could reach more than 1mD for a large mercury injected volume, while the permeability was mostly distributed between 0.1-1mD for a low percentage of injected mercury.3) High displacement pressure classification group Type 11–15 results showed that the most important

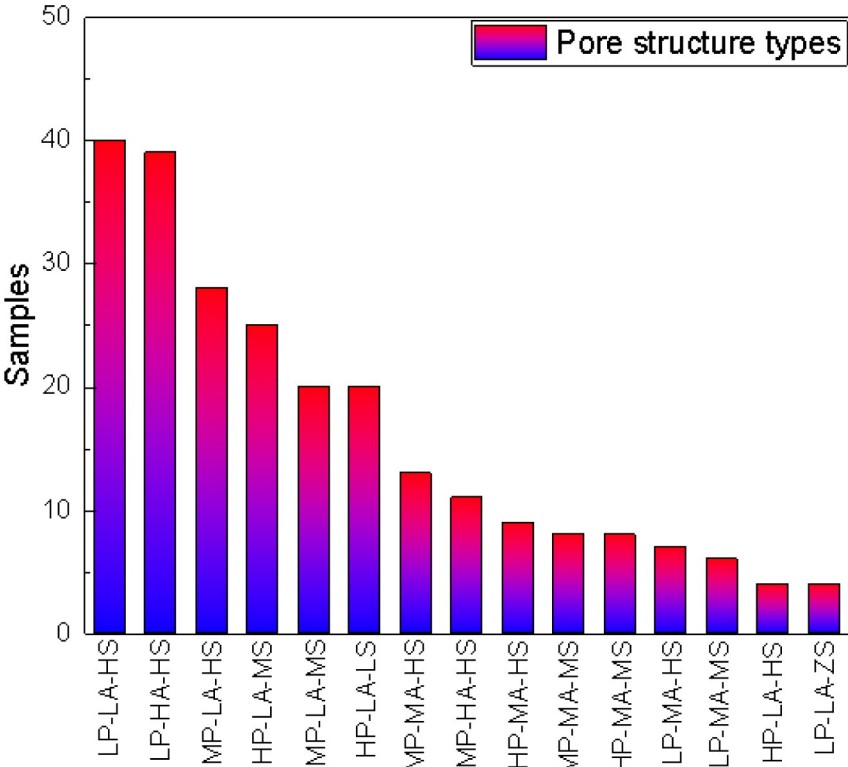

**Fig 6. Statistics of pore structure types on MICP-based morphological classification.**

reason for the high displacement pressure was the absence of connected pore-throat >0.63μm. If the sorting is middle, there is a certain proportion of 0.25–0.63μm pore throat. Except for the worst two types of HP-LA-MS and HP-LA-LS with the median value below 7%, the remaining three typeshave porosity values still at 10%, and the permeability distribution is in the range of 0.1-1mD.

On the whole, except for the features reflected by samples of extreme pore structure types, the other transition types show the similarity of features. The displacement pressure is closely related to the type of dominant pore-throat, the sorting is closely related to the proportion of dominant pore-throat, and the percentage of mercury injected is closely related to the distribution of porosity. The homogenization of porosity and the complexity of permeability are the main macroscopic characteristics of the complex pore structure.

## 4.3 Data-mining analysis based on pre-classification scheme

**4.3.1 Quantitative parameter abstracted and pre-analysis.** There are many quantitative characterization parameters of pore structure. Some scholars have extracted 48 kinds of quantitative characterization parameters for pore structure characterization. However, having too many characterization parameters can lead to overlapping information, resulting in strong collinearity and inevitably improved classification effect. Based on previous studies [35, 36], the characterization parameters with large differences in pore structure characteristics of different categories were analyzed and studied. Porosity, permeability, median pressure, displacement pressure, relative sorting coefficient, microscopic mean coefficient, mean pore-throat

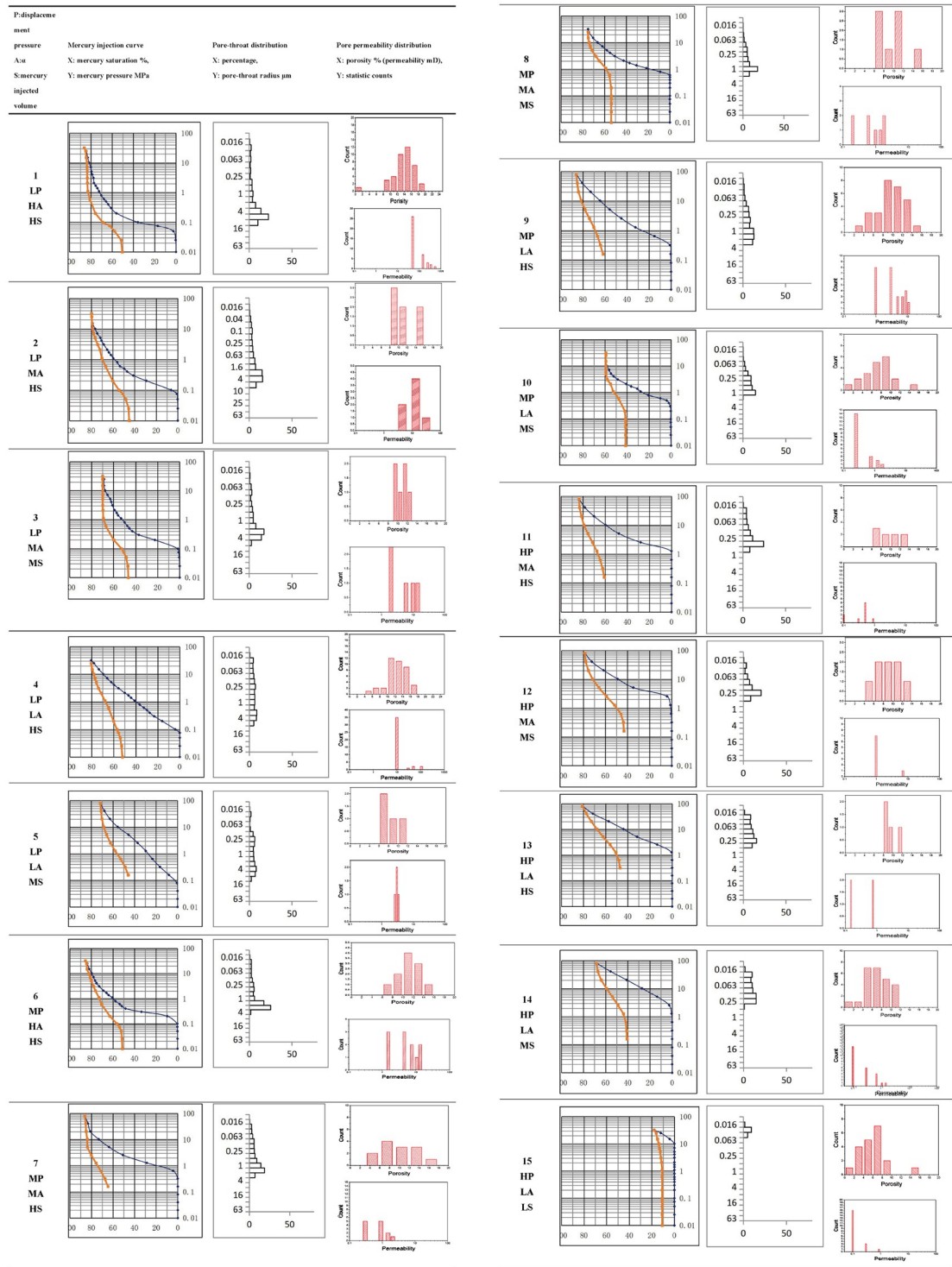

**Fig 7. Morphological MICP classification and histogram of porosity and permeability.**

diameter, maximum pore-throat radius, mean pore-throat radius and median pore-throat radius were selected to quantitatively characterize the pore structure.

The pore structure parameters extracted from mercury injection data were used to conduct pairwise intersection of structural characterization parameters for pairplot plotting of characteristic parameters, as shown in Fig 8:

According to pairplot, among the characteristic parameters of mercury injection, maximum pore radius, median pore radius, average pore radius, median pressure and relative sorting coefficient are all significantly correlated with permeability. These pore structure parameters not only reflect on pore throat size but also on the influence of pore-throat distribution and circuitous situation on percolation. The pore structure with high displacement pressure has the characteristics of low mean pore-throat radius, low maximum pore-throat radius and high median pressure. However, the parameters overlap more seriously, and the permeability also overlaps more seriously with the middle displacement pressure type. The pore structure with medium displacement pressure has the characteristics of medium pore-throat radius average, medium pore-throat radius median. The parameters overlap more significantly, and the permeability overlap with the high displacement type is also more significant. The low displacement pore structure has high mean pore-throat radius, high maximum

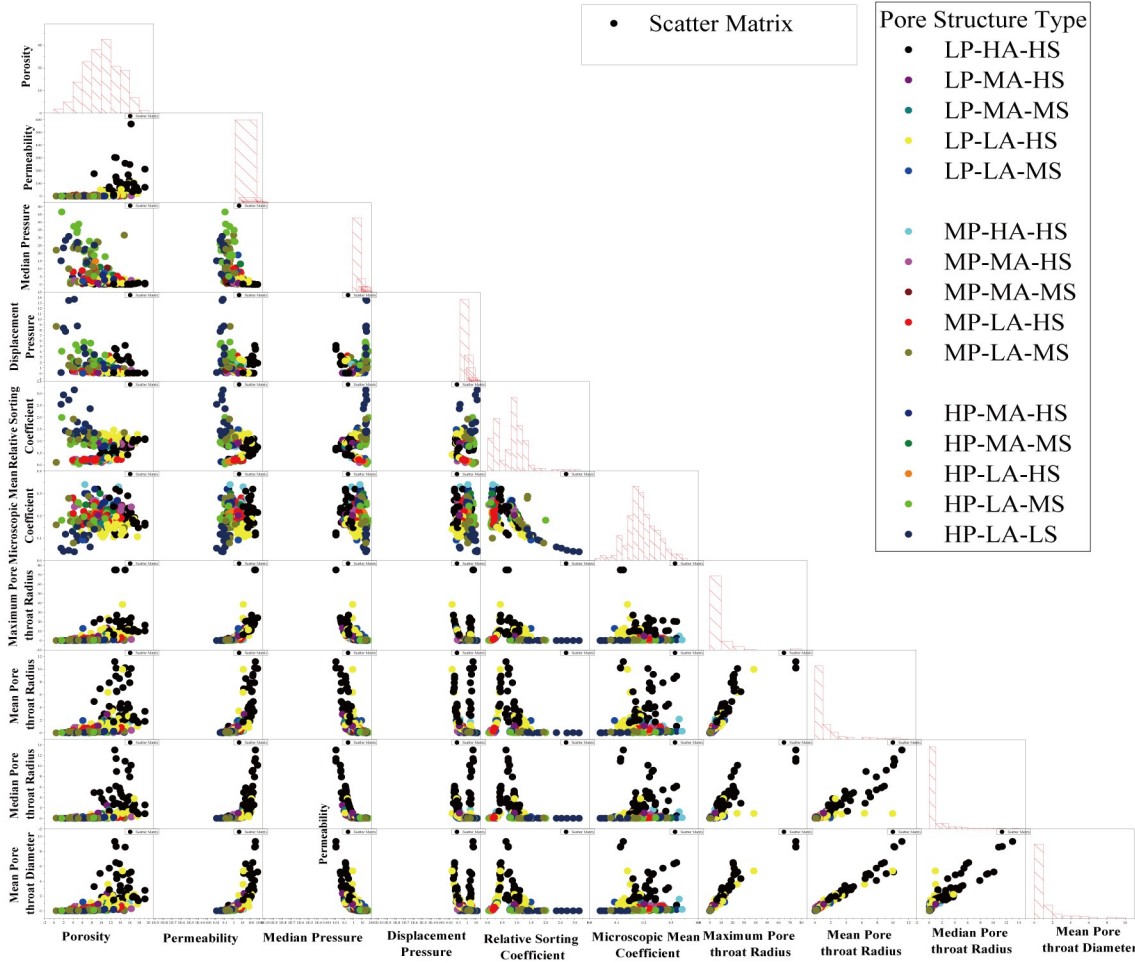

**Fig 8. Pairplot of characteristic parameters of MICP data.**

pore-throat radius, and high median pore-throat radius. The low-displacement pressure pore structure has the characteristics of partial overlapping distribution in the other parameter distributions, and the percolation performance of the low-displacement pore structure is the best, and it has the obvious characteristics of medium and high permeability.

**4.3.2 Compression and analysis of characteristic parameters of MICP.** The following figure shows the cluster dendrogram calculated between screened variables using average-linkage algorithm as the inter-cluster distance algorithm and Pearson correlation coefficient as the connection metric, as shown in Fig 9. In this figure, the clustering process is represented in an intuitive way. The dendrogram counts the maximum distance between variables as 25, and the remaining distances are converted to relative distances.

Decision tree is a tool to represent logical processing by binary tree graph. In the decision tree algorithm, the classification weight of the root node is the highest and decreases in descending order. Selecting the feature parameter with the strongest classification ability as the root node can greatly improve the classification efficiency. The classification ability of each feature is quantified by the information gain. The greater the information gain of the feature, the stronger the classification ability. The information gain of each feature point in the data set is calculated, and the feature point with the largest information gain is used as the decision root node, which is recursed downward successively. Since the root node can quantify the feature parameters, this method is often used to evaluate the feature parameters importance. The decision tree based on mercury injection characterization parameters is shown in Fig 10.

Combined with Figs 9 and 10, the results of the relationship between variables calculated by hierarchical clustering method correspond to the results run by decision tree to a certain extent, and the following information can be obtained: 1) The screening variables can be divided into three groups: Group1:mean radius-maximum radius-median radius-pore-throat diameter mean variable group, Group2:microscopic mean coefficient variable group; Group3:

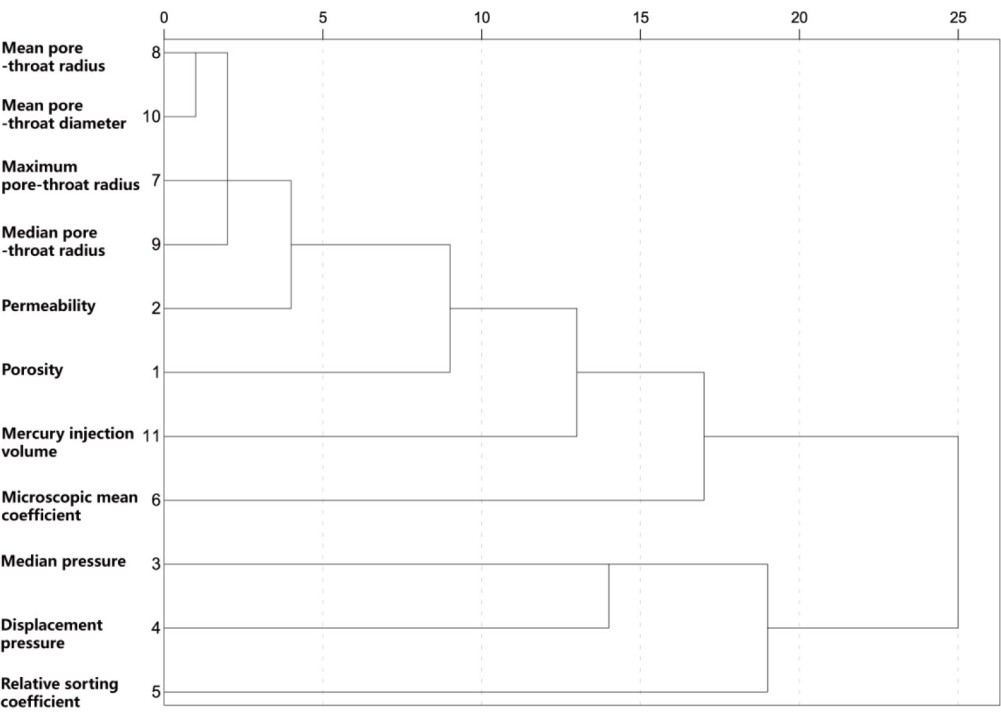

**Fig 9. Hierarchical cluster diagram based on characteristic parameters of MICP.**

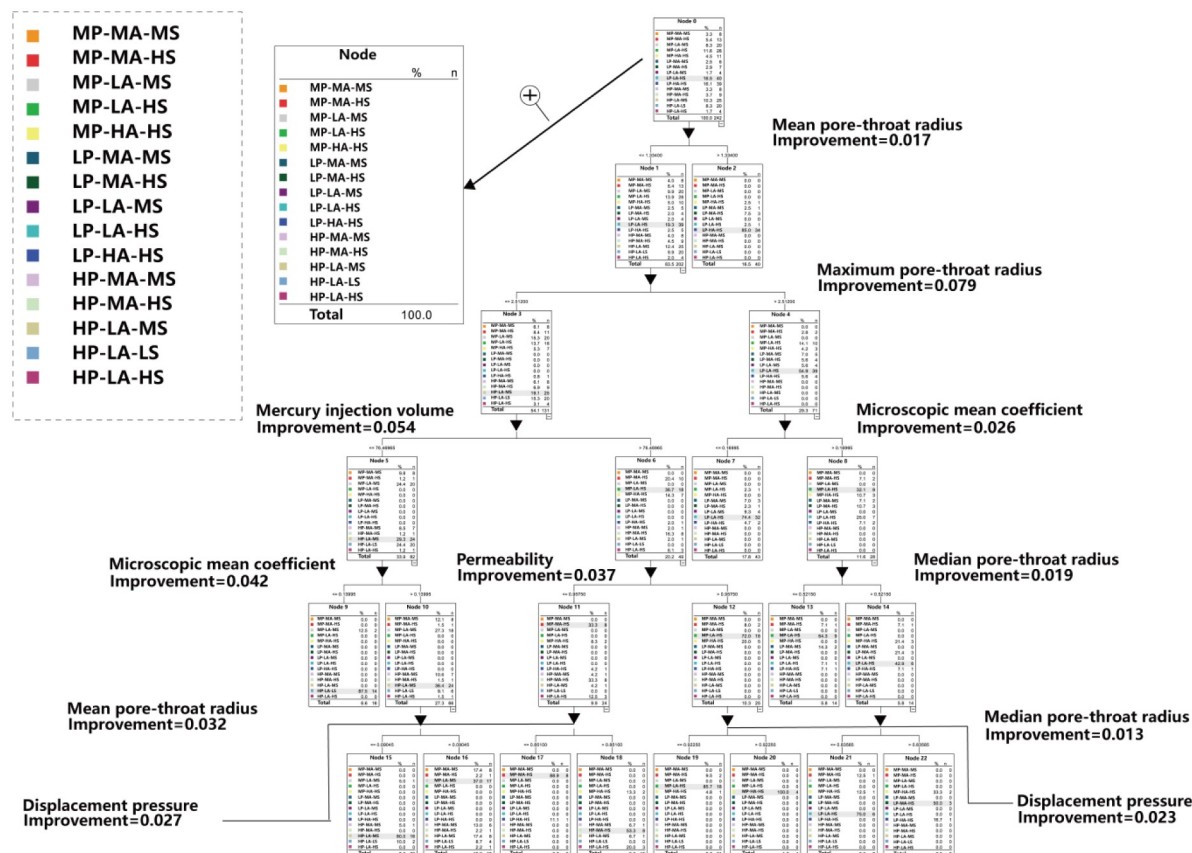

**Fig 10. Diagram of decision tree based on characteristic parameters of MICP.**

median pressure-displacement pressure-relative sorting coefficient variable group. 2) The pore-throat radius group is close to the permeability in dimension, and the first and second nodes of the corresponding decision tree are the pore-throat radius group. The two methods are consistent in the concept that the pore-throat radius plays a decisive role in pore structure. 3) The microscopic mean coefficient group has only one parameter, which is relatively independent in the information dimension, and it is located in the third-layer node of the decision tree. 4) Although the median pressure, displacement pressure and relative sorting coefficient are all in the same group, the distance within the group is large, so the information collinearity is low and the information is relatively independent.

Based on hierarchical clustering analysis and decision tree analysis above, we took the microscopic mean coefficient, median pressure, displacement pressure and relative sorting coefficient as independent parameters, and compressed the mean variable group with strong collinearity in components, namely the mean pore-throat radius-maximum pore-throat radius-median pore-throat radius-pore-throat diameter. Principal component analysis was used to extract the number of principal components according to the statistical standard, that is, the default standard with characteristic root greater than 1. The number of extracted principal components is determined synthetically according to the practical problems combined with the scree plot.

Scree Plot can display the importance degree of each factor. The horizontal axis is the number of the factor and the vertical axis is the number of the characteristic root. The factors are

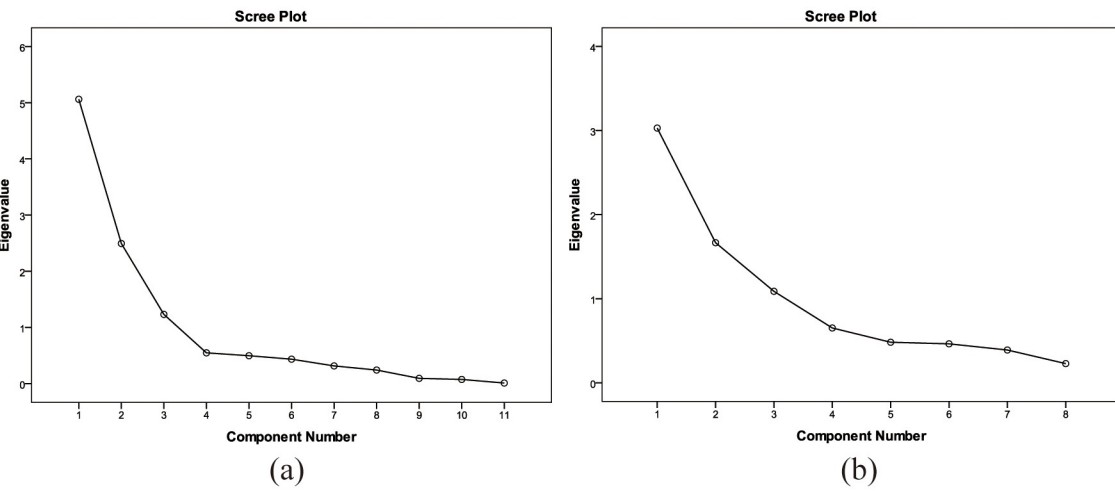

**Fig 11. Scree plot of the original dataset and after compressed.**

arranged in order from the largest to the smallest characteristic root, and the main factors can be directly observed from the Scree Plot. Steep slope corresponds to large characteristic roots, while platform corresponds to small characteristic roots with weak influence. Therefore, for variables with relatively independent information from each other, the slope is required to be consistent, while for variables with strong collinearity, the slope change is obviously from steep to slow. The scree plot comparison of the dataset between original and after compressed are shown in Fig 11.

As can be seen from Fig 11(a), the components tend to be slope steep showing a great collinearity; therefore, the compression of variable group into a comprehensive parameter is the compression Factor 1. We combined compression Factor 1, porosity, permeability, microscopic mean coefficient, median pressure, displacement pressure, relative sorting coefficient and mercury percentage to Plot the Scree Plot. As can be seen from Fig 11(b), the overall slope is stable, indicating that the variables are relatively independent and the pore structure can be characterized in different dimensions, and the variable group can be used to characterize the pore structure parameters.

## 4.4 Merged classification scheme based on LDA model and hierarchical clustering model

Since the pre-classification scheme classifies the pore structure into 15 types, which can be separated in the dimension of mercury injection characteristic parameters, but may not have obvious characteristics in response and hard to utilize in application, there is a need to integrate the categories. In the subsequent work, we used the Linear Discriminant Analysis (LDA) method combined with the hierarchical clustering method [37, 38] based on morphological pre-classification scheme to analyze variable groups, so as to provide the proof for the combined classification scheme.

Linear discriminant model (LDA) is a dimensionality reduction technique of supervised learning, which uses the pre-classification scheme as the supervised sample to conduct the center analysis of different types. We should project data on low dimensions from high dimensions. We created a two-dimensional projection crossplot based on the LDA model, as shown in Fig 12.

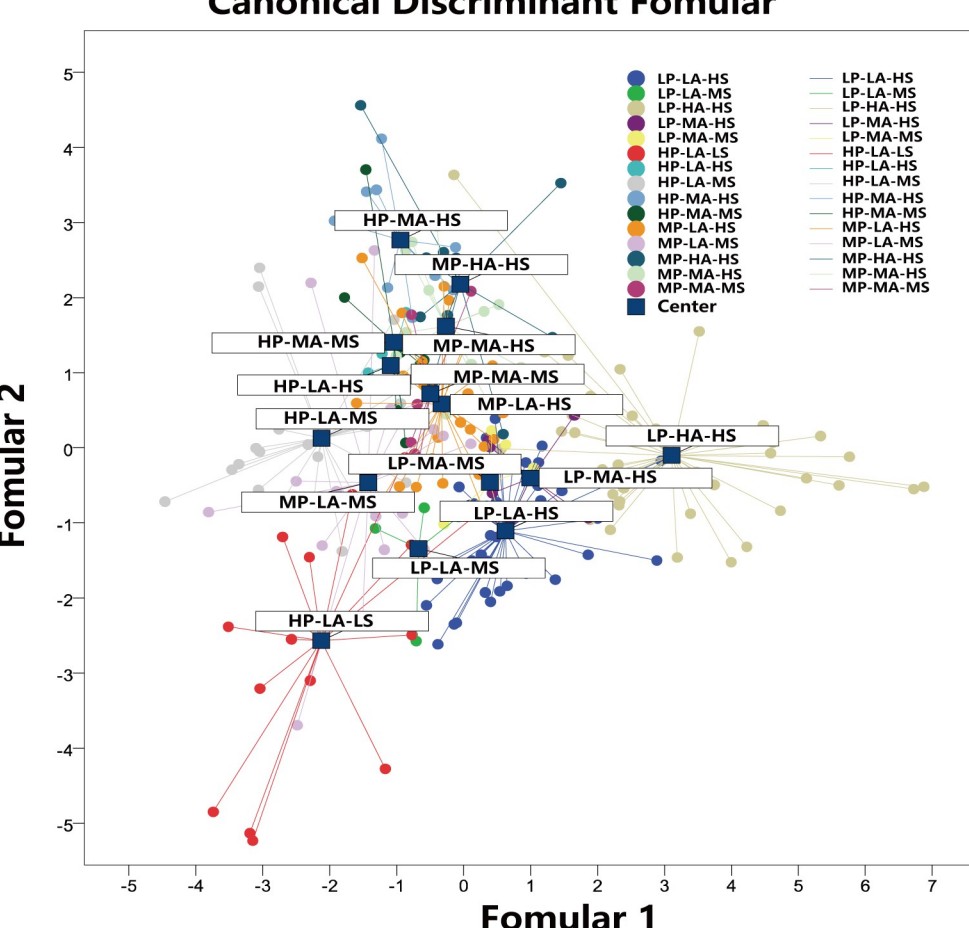

**Fig 12. Projection crossplot of linear discriminant model.**

We hope that the projection of data of the same type should be as close as possible, and the distance between the category centers of data of different types should be as large as possible. The LDA model projects the pore structure samples in the high-dimensional space to the low-dimensional space, making the projected sample data have the minimum intra-class distance and the maximum inter-class distance in the new subspace, making the best separability in the subspace, so as to confirm the distance between different types of pore structures in the low-dimensional space where the information is concentrated. It is used to recognize the relationship between different types and provide integrated evidence. According to the knowledge above, it can be seen in Fig 12: 1) the changes of displacement pressure information dimension mainly concentrated in the upper left—lower right direction. The displacement pressure of the same level is distributed along the direction of the lower left and the upper right, so three distribution lines of low displacement pressure, middle displacement pressure and high displacement pressure can be obtained, which are nearly parallel. The change in the direction of lower left and upper right is mainly the change of sorting, showing a change from low to high. 2) The smaller inter-class distance is mainly concentrated in different displacement pressure levels. Low sorting and high injected volume are similar to mid sorting mid injected volume. Different displacement pressure levels show the same nature, so it is very important to understand

**Table 3. Pre-classified pore structure parameters median based on distribution.**

| Types<br>Parameters | 1 | 2 | 3 | 4 | 5 | 6 | 7 | 8 | 9 | 10 | 11 | 12 | 13 | 14 | 15 |
|---|---|---|---|---|---|---|---|---|---|---|---|---|---|---|---|
| Porosity, % | 14.11 | 11.20 | 10.55 | 12.50 | 8.00 | 11.58 | 10.02 | 9.62 | 9.29 | 7.85 | 9.60 | 9.16 | 9.12 | 7.40 | 6.00 |
| Permeability, Md | 67.99 | 13.70 | 4.65 | 6.82 | 3.11 | 5.68 | 0.68 | 0.76 | 3.00 | 0.32 | 0.43 | 0.44 | 0.43 | 0.18 | 0.08 |
| Median pressure, MPa | 0.11 | 0.13 | 1.99 | 1.67 | 7.24 | 0.92 | 2.52 | 2.91 | 3.63 | 8.62 | 5.11 | 6.32 | 8.65 | 11.69 | 18.56 |
| Displacement pressure, MPa | 0.08 | 0.11 | 0.13 | 0.15 | 0.10 | 1.11 | 0.32 | 0.71 | 0.32 | 1.38 | 1.22 | 1.24 | 1.26 | 1.17 | 1.69 |
| Relative sorting coefficient | 0.72 | 0.91 | 0.85 | 1.08 | 0.28 | 0.29 | 0.23 | 0.92 | 0.23 | 1.08 | 0.77 | 0.92 | 0.49 | 0.93 | 1.41 |
| Microscopic mean coefficient | 0.21 | 0.18 | 0.18 | 0.15 | 0.15 | 0.26 | 0.24 | 0.21 | 0.20 | 0.17 | 0.27 | 0.22 | 0.22 | 0.18 | 0.12 |
| Mercury injection volume% | 94 | 91 | 83 | 90 | 74 | 91 | 93 | 82 | 91 | 71 | 93 | 83 | 91 | 70 | 25 |
| Compressed factor 1 | 6.46 | 2.57 | 2.60 | 2.31 | 2.19 | 0.61 | 0.70 | 0.58 | 1.05 | 0.29 | 0.33 | 0.25 | 0.28 | 0.15 | 0.9 |

the geological law of this kind of phenomenon. 3) The position of high displacement, low sorting and low inflow mercury in this figure is relatively special. It is far away from all categories, and far away from the high displacement pressure level group. Therefore, it exists as a special type of outlier, and the biggest possibility is that the sample properties have fundamentally changed.

Based on the pre-classification scheme as the analysis dimension, and the median value of pore structure parameters of each category was taken as the variable to conduct hierarchical clustering analysis on the relationship between each type, as shown in Table 3 and Fig 13.

According to the results, the pore structure of Type 1 and Type 5 has obvious independent characteristics of pore structure parameters, and the relative distance of hierarchical clustering

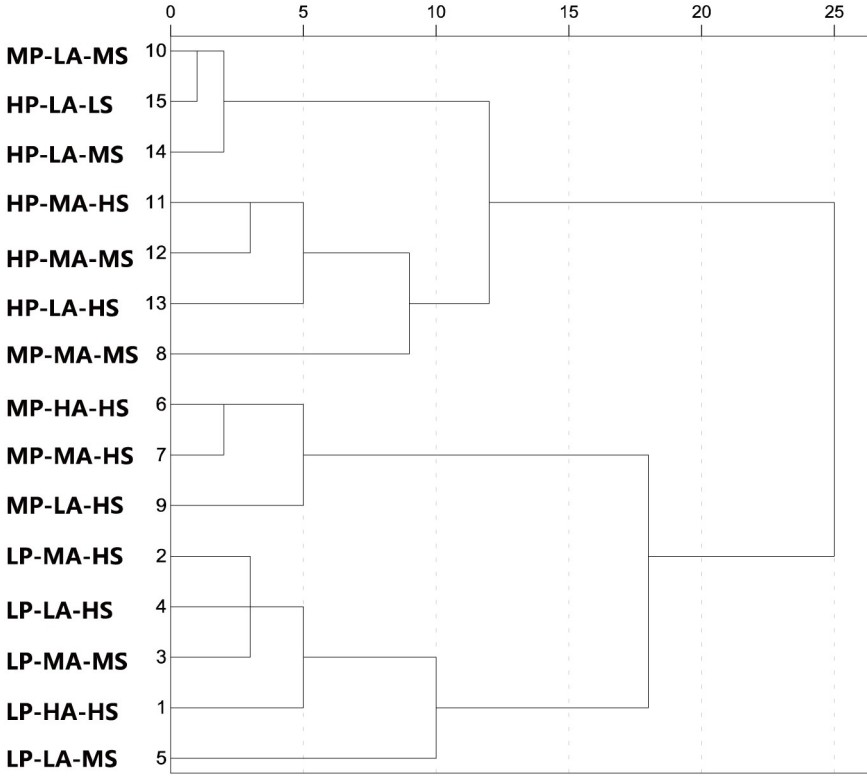

**Fig 13. Hierarchical diagram of pore structure categories based on morphological pre-classification.**

is greater than 5. Corresponding to the LDA projection, the cluster distance is far away from the other clusters. Therefore, it can be shown that the characteristics of mercury injection parameters are obvious, and this category is relatively significant as a single category and the pore structure is the most developed pore-throat network. The pore structures of Type 2, 3, and 4 are close to each other, and the distance between groups is smaller by hierarchical clustering analysis, the relative distance is less than 5, which reflects more of the pore types that exist at the same time with a certain degree of transformation and blockage of pore throat. Type 6 and Type 7 show similar properties. The distance between LDA projections is relatively close, and the distance between groups is also small in cluster analysis, which indicates the existence of pore structure as a transitional type. The distribution of LDA projections of Type 8, 12, and 13 is very similar, and they mostly reflect complex pore structures with high displacement pressure and high mercury injected volume. The distribution of LDA projection maps of Type 10, Type 14 and type 15 are all located in the lower left region, and the relative distance between the groups is only 2.3 according to the cluster analysis, so they are merged into a type, which reflects the lowest pore structure. Type 11 shows a certain degree of contradiction in the LDA projection diagram and hierarchical clustering analysis diagram. Type 11 is distributed independently in the LDA projection diagram and is located above the diagram, but it is far away from the types in the hierarchical diagram. Similarly, Type 9 is the closest to type 8 in the LDA projection, but it is not in the same branch with Type 8 in hierarchical clustering. Finally, the hierarchical clustering analysis method based on entropy gain calculation was determined as the standard, and the Type 9 was merged into the same group with the Type 6 and 7, and the Type 11 was merged into the same group as the Type 8, 12, 13. The final pore structure merging scheme is shown in Fig 14.

## 4.5 Calibration by core observations of the final pore type scheme

Based on the mercury injection data, the pore structure types were tested and calibrated by core observations such as thin sections, cast thin sections, cathode luminescence and scanning electron microscopy. The characteristics of different types of pore structure are shown in Fig 15.

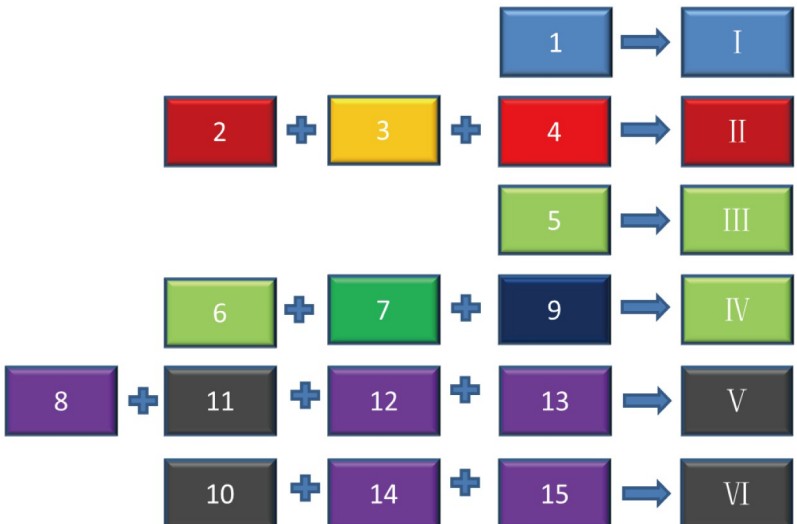

**Fig 14. Final pore structure classification scheme.**

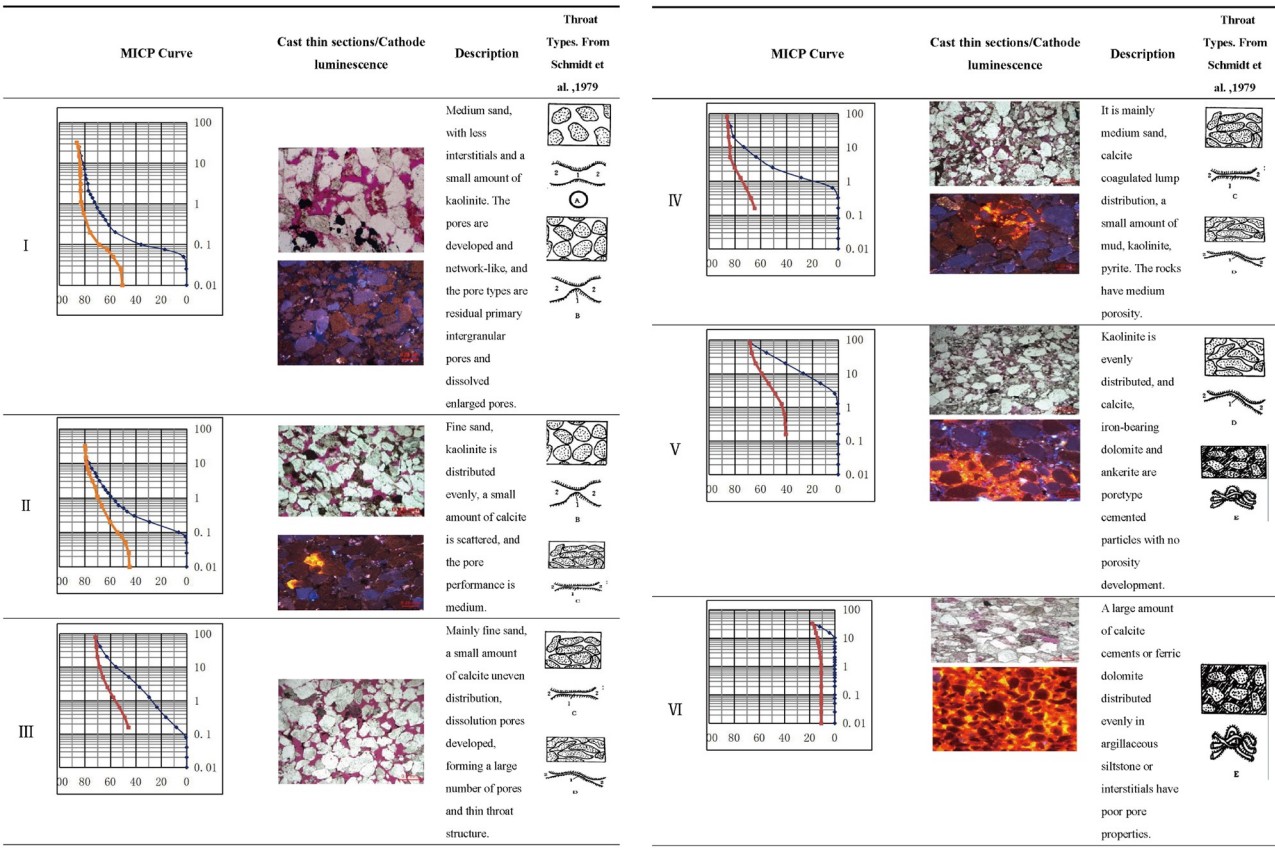

**Fig 15. Pore structure classification and microscopic characteristics.**

Cathode luminescence indicates the components with different colors [6, 39] with quartz brown and blue-purple light, quartz secondary edge does not emit light, quartzite debris brown, feldspar blue light, rock debris brown light, kaolinite complex group indigo light, calcite cement orange yellow, orange red light, iron dolomite does not emit light, iron dolomite orange red light. The throat types is as followed [35, 40]: A, pore reduction throat; B, Constricted throat; C, Flaky throat; D, Curved flaky throat; E, Tubular bundle throat.

Type I has lower Pd, larger pore throat median radius and better sorting performance. It is mostly fine-lithic quartz sandstone of PS2 or PS3 submember, with little feldspar content, a small portion of clay minerals distribution, and basically no carbonate cements. The thin sections show that the reservoir space is dominated by residual primary intergranular pores, with a small amount of secondary pores formed by the dissolution of feldspar, and a high content of authentic quartz. The high content of quartz particles can prevent the secondary increase of quartz, and the dissolution of feldspar and rock debris can increase the pore volume and improve the pore structure, so it is the best reservoir in the study area.

Type III is characterized by low Pd, large median pore-throat radius and good pore-throat sorting. Most of them are lithic quartz sandstone in PS3 and PS4, with a certain content of clay minerals and a small amount of carbonate cements. The pore types are also dominated by residual primary intergranular pores and secondary pores, and the secondary pores are mainly feldspar dissolution pores with high primary pore content. The clay minerals with uniform distribution (mainly kaolinite) on the surface of quartz grains reduce the volume of some meso-

macropores, increase the proportion of thin throats, and reduce the sorting ability of the reservoir.

Type III has low Pd, pore-throat median radius is small, pore-throat sorting is poor, and most of them are lithic quartz siltstone in PS5. The reservoir space is characterized by large pores and thin throats, and the reservoir space is mainly caused by the strong dissolution of carbonate cementation. The main reservoir space is secondary dissolution pore, which is a unique type of multi-cement and multi-dissolution pore formed by local dissolution of some debris and calcite cements.

Type IV has high Pd, large pore-throat radius and general sorting property. It is medium-fine lithic quartz sandstone of PS4. The content of feldspar and clay (kaolinite) is high, and the content of calcite and ankerite cements is mostly 5–10%. The core thin section shows that the reservoir space is mainly residual primary intergranular pores, and the secondary pores formed by the dissolution of feldspar are also relatively developed. Carbonate is filling pore between particles, before the start of the cementation, and it needs to have connectivity good pore system development, to allow enough rich carbonate fluid into the sand body, and because of the formation water in the sand body with change, carbonate precipitation, and the embedded in crystalline cemented sand body occupy large hole, displacement pressure increased significantly. At the same time, the pore structure of interconnected small holes and small roar is developed.

Type V is medium-fine lithic quartz sandstone of PS4-PS5 member with higher Pd, smaller pore-throat radius and poor sorting performance. The clay content is high, and the calcite and ankerite cements are developed, the content of which is more than 10%. The thin section analysis shows that the reservoir space is mainly composed of residual primary intergranular pores and micropores formed by the dissolution of debris or impurity. This type of pore structure is mostly developed in argillaceous siltstone, which is reflected as a pore type formed by the inlaid crystal filling of the late developed cements, and the displacement pressure increases obviously.

Type VI has highest Pd, extremely small pore throat radius, very poor sorting, and low porosity and permeability. The reservoir space is mainly intergranular micropores, and most of them are tight layers. The samples are mostly fine siltstone of mucocalcium or fine sandstone with strong carbonate cementation, and the dissolution is weak.

## 5. Conclusion

In this paper, we conducted a data-mining-based pipeline on 314 MICP samples to investigate the characterization and classification of the strongly diagenetic sandstones in the Upper 3rd submember of Kepingtag.

The following conclusions are:

1.  The pore structure was preclassified according to the morphological classification criteria. 314 samples were classified into 15 pore structure types according to displacement pressure (Pd), Smin and α angle. The sample on pre-classification by morphology shows 15 types. According to the statistics of different types of samples, LP-LA-HS and LP-HA-HS were the most developed.

2.  The displacement pressure is closely related to the type of dominant pore-throat: the main reason for the low displacement pressure was the existence of connected pore-throats >4μm; The high displacement pressure was due to the absence of connected pore-throat >0.63μm. The sorting is closely related to the proportion of dominant pore-throat, and the percentage of mercury injected is closely related to the distribution of porosity. The

homogenization of porosity and the complexity of permeability are the main macroscopic characteristics of the complex pore structure.

3. Porosity, permeability, median pressure, displacement pressure, relative sorting coefficient, microscopic mean coefficient, mean pore-throat diameter, maximum pore-throat radius, mean pore-throat radius and median pore-throat radius were selected to quantitatively characterize the pore structure. According to analysis of hierarchical clustering and decision tree, the screening variables can be separated into three groups: Group1: mean radius-maximum radius-median radius-pore-throat diameter mean variable group; Group2: microscopic mean coefficient variable group; Group3: median pressure-displacement pressure-relative sorting coefficient variable group. Combined with PCA analysis, the compression of variable group 1 into a comprehensive parameter is the compression Factor 1.

4. The pore structure final classification scheme is verified by LDA analysis and hierarchical clustering based on data-mining. In LDA projection crossplot, we found the changes of displacement pressure information dimension mainly concentrated in the upper left—lower right direction and three distribution lines are discovered which are nearly parallel. The change in the direction of lower left and upper right is mainly the change of sorting, showing a change from low to high. Combined with results on hierarchical clustering, we establish a final scheme on 6 types.

5. The final pore types are calibrated on core observation. Type I has mostly fine-lithic quartz sandstone of PS2 or PS3 submember. Type II is characterized by lithic quartz sandstone in PS3 and PS4, with a certain content of clay minerals and a small amount of carbonate cements. Type III are lithic quartz siltstone in PS5. Type IV is medium-fine lithic quartz sandstone of PS4. Type V is medium-fine lithic quartz sandstone of PS4-PS5 member. Type VI are mostly fine siltstone of mucocalcium or fine sandstone with strong carbonate cementation, and the dissolution is weak.

The paper employs a data-mining based pipeline to analyze samples for the characterization and classification of strongly diagenetic sandstones. This indicates an innovative approach to leverage data mining techniques in geoscience and petroleum engineering for understanding subsurface rock properties. It introduces a comprehensive set of quantitative parameters to characterize in combination with the use of data-mining analytics. This approach provides a more nuanced understanding of the complex pore structure, going beyond traditional porosity and permeability measurements. This paper presents a novel classification method that integrates data mining with geological knowledge. By applying data mining techniques to analyze mercury intrusion data and other datasets, we uncovered and classified the hidden information of pore structures. This innovative approach significantly enhances classification accuracy and minimizes the influence of human factors, fully harnessing the strengths of data mining. Our method introduces a groundbreaking technique for pore structure classification, underscoring its innovation and importance in the field.

## Supporting information

**S1 Table. Dataset for pore type classification.** This supplementary file contains the dataset consisting of all 314 samples, encompassing a diverse range of attributes parameter, relevant for classification of the pore types.
(XLSX)

## Acknowledgments

I am thankful to Meiling Zhang reviewing for comments and improvement to this manuscript. Comments from others reviewers, Berdan and Seuhan is acknowledged.

## Author Contributions

**Conceptualization:** Xidong Wang.

**Data curation:** Feng Tian, Xidong Wang, Xinyi Yuan.

**Formal analysis:** Xidong Wang.

**Funding acquisition:** Xinyi Yuan.

**Investigation:** Xidong Wang.

**Methodology:** Feng Tian, Xidong Wang, Di Wang.

**Project administration:** Xinyi Yuan, Di Wang.

**Software:** Di Wang.

**Supervision:** Xinyi Yuan.

**Validation:** Xidong Wang.

**Visualization:** Feng Tian, Xidong Wang, Di Wang.

**Writing – original draft:** Feng Tian, Xidong Wang.

**Writing – review & editing:** Feng Tian, Xidong Wang.

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
