## [Decision Letter · Decision Letter 0]

17 Nov 2023

PONE-D-23-28579Improved Pore Structure Characterization and Classification of Strong Diagenesis Sandstones by Data-mining Analytics in Tazhong Area, Tarim Basin

PLOS ONE

Dear Dr. Wang,

Thank you for submitting your manuscript to PLOS ONE. After careful consideration, we feel that it has merit but does not fully meet PLOS ONE’s publication criteria as it currently stands. Therefore, we invite you to submit a revised version of the manuscript that addresses the points raised during the review process.

Besides considering all the comments by the reviewers, I advise to pay close attention to the following topics:

Please separated methodology and results. On the other hand, analysis, testing methods and quantities can be combined in a single section.Please optimize the Tables and Figures of the manuscript.Please add a more detailed description of the new findings and significance of the paper. This may require additional references to better situate the reader with respect to the state-of-the-art.

We look forward to receiving your revised manuscript.

Kind regards,

Luan Carlos de Sena Monteiro Ozelim, D.Sc.

Academic Editor

PLOS ONE

Journal Requirements:

2. Did you know that depositing data in a repository is associated with up to a 25% citation advantage (https://doi.org/10.1371/journal.pone.0230416)? If you’ve not already done so, consider depositing your raw data in a repository to ensure your work is read, appreciated and cited by the largest possible audience. You’ll also earn an Accessible Data icon on your published paper if you deposit your data in any participating repository.

Tianchi talent project (Grant No.40300-23005104)

I am thankful to Meiling Zhang reviewing for comments and improvement to this manuscript. Comments from others reviewers, Berdan and Seuhan is acknowledged. This research is jointly funded by Tianchi project (Grant No.ZD2019-183-006).

Tianchi talent project (Grant No.40300-23005104)

Additional Editor Comments:

Dear Authors,

After careful analysis by two experts, to further proceed with the publication of the paper, a Major Review is required.

Besides considering all the comments by the reviewers, I advise to pay close attention to the following topics:

- Please separated methodology and results. On the other hand, analysis, testing methods and quantities can be combined in a single section.

- Please optimize the Tables and Figures of the manuscript.

- Please add a more detailed description of the new findings and significance of the paper. This may require additional references to better situate the reader with respect to the state-of-the-art.

Reviewers' comments:

Reviewer's Responses to Questions

**Comments to the Author**

1. Is the manuscript technically sound, and do the data support the conclusions?

Reviewer #1: Yes

Reviewer #2: Yes

2. Has the statistical analysis been performed appropriately and rigorously? 

Reviewer #1: Yes

Reviewer #2: Yes

3. Have the authors made all data underlying the findings in their manuscript fully available?

Reviewer #1: Yes

Reviewer #2: Yes

4. Is the manuscript presented in an intelligible fashion and written in standard English?

Reviewer #1: Yes

Reviewer #2: Yes

5. Review Comments to the Author

Reviewer #1: The manuscript is about the pore structure characteristics of sandstones in Tazhong area.

The text provides some new insights. However, this manuscript cannot be accepted in present state, unless the following issues and mistakes are modified. Major revision recommended.

1. Petrology characteristics of sandstone should be moved to the results part, it is your own research results, right? And the sample number of sandstone should be added in Fig.2 and Fig.3.

2. A and B of Fig, 3 did not include Folk (1973) sandstone classification, what’s the figure caption of c and d, which should include sample data, depth, PPL or XPL or CL.

3. Line 84, X-ray diffraction mentioned, where is your XRD data?

4. The author combined the methodology and results together, it is confusing. The analysis and testing methods and quantities should be put together.

5. The whole manuscript is actually based on the classification of sandstones using thin sections and mercury intrusion, but I donot understand the roles of ‘strong diagenesis’ in it, what’s impact of strong diagenesis on the formation of sandstone.

6. What is MICP? I did not see the full name of it in the text.

7. What is the function of cathode luminescence in the pore structure classification?

Reviewer #2: The manuscript attempts to establish a framework for pore structure characterization and classification of strong diagenesis sandstones, combined with data analysis methods. Overall, this paper has certain innovation and practical significance.

The following comments and suggestions should guide the authors to better revise the paper:

1.The abstract of the manuscript is very detailed. However, sometimes such abstract cannot clearly reflect the key insights and particularities of the research. Therefore, it is suggested that the authors optimize the structure of the abstract to make it more concise.

2.References should be supplemented with the latest research results. The introduction section lacks support of recent years. It is suggested that authors update references. Due to the lack of support from relevant references, it is impossible to determine whether the author has tracked the latest progress, which may lead to problems in summarizing the current situation and sorting out the problems. It is very important for readers to understand the research background.

3.The text in Figure 1 should be in Times New Roman font. This Figure can be further optimized. The scale in Figure 3 cannot be seen clearly. Part of the text in Figure 6 is too small to read. Fig.8 is not clear, and the content cannot be seen clearly. Some text sizes in Figure 10 exceed the main text. They're lacks of explaining the Figure 11. please describe more.

4.Tables 2 and 5 have inserted figures in the table. Perhaps it would be more appropriate to directly modify it to a highly summarized figure. The authors can consider whether modifications are needed (this is only a suggestion)

5.Please further revise the statements in the paper that are difficult to read. Especially the naming of some new classifications. Part of the paper's discussion is not very user-friendly to readers and can be further optimized for expression. Commonly the abbreviations are explained the first time of the appearance of the corresponding terms. Here the abbreviations MICP, PCA, LDA, appearing in abstract, should be attributed to specific terms.

6.Please add more description of the new findings and significance of research. Especially in the abstract and conclusion section.

6. PLOS authors have the option to publish the peer review history of their article (what does this mean?). If published, this will include your full peer review and any attached files.

Reviewer #1: No

Reviewer #2: No

---

## [Author Response · Author response to Decision Letter 0]

30 Dec 2023

Dear Editors and Reviewers:

Thank you for your letter and for the reviewer comments concerning our manuscript entitled “Improved Pore Structure Characterization and Classification of Strong Diagenesis Sandstones by Data-mining Analytics in Tazhong Area, Tarim Basin”. Those comments are all valuable and very helpful for revising and improving our paper, as well as the important guiding significance to our successive researches. Comments are carefully studied and have made correction which we hope meet with approval. Revised portion are marked in yellow in the paper. 

The main corrections in the paper and the responds to the reviewer’s comments are as following:

Responds to the reviewer’s comments:

Reviewer #1: 

Thanks for your comments on our paper very much. It is a great encouragement to our research. The majority of manuscript is revised according by your complete comments and your comments bring us lot thinking on further research. The revisions are mainly focused on following aspects.

1. Response to comment: Petrology characteristics of sandstone should be moved to the results part, it is your own research results, right? And the sample number of sandstone should be added in Fig.2 and Fig.3.

Response: 

Thank you for pointing out the oversight. As the reviewer's valuable suggestion, we have made the necessary adjustments to our manuscript. The petrographic characteristics of the sandstone samples have been relocated to the Results section, where they are more appropriately discussed. Additionally, we have included the number of sandstone samples in both Figure 2 and Figure 3, ensuring a more comprehensive presentation of our findings. Thank you for the suggestion.

2. A and B of Fig, 3 did not include Folk (1973) sandstone classification, what’s the figure captions of c and d, which should include sample data, depth, PPL or XPL or CL.

Response: 

We appreciate the reviewer's feedback and have made the necessary revisions to the citation of A and B of Fig, 4. Additionally, we have changed the name of Figure 3 accordingly. Thank you for the suggestion.

3. Line 84, X-ray diffraction mentioned, where is your XRD data?

Response：

It has been published in previous study in Journal of Applied Geophysics, so we had added in the reference. 

4. Response to comment: The author combined the methodology and results together, it is confusing. The analysis and testing methods and quantities should be put together.

Response: 

As your recommendation, we have now separated the research methodology and results sections, allowing for a clearer distinction between the two. Thank you for the suggestion.

5. Response to comment: The whole manuscript is actually based on the classification of sandstones using thin sections and mercury intrusion, but I do not understand the roles of ‘strong diagenesis’ in it, what’s impact of strong diagenesis on the formation of sandstone.

Response:

We have made the necessary additions in our manuscript, specifically on line 359-396, and in Table 5, it illustrated that for Type Ⅲ to Ⅵ, it reflects strong diagenesis effect on the pore structure, Where we have emphasized the significant role of diagenesis in the formation process of sandstone. Thank you for the suggestion.

6. Response to comment: What is MICP? I did not see the full name of it in the text.

Response：

In accordance with the reviewer's suggestion, we have now included the full names of the abbreviations upon their first occurrence in the text and we have compiled all abbreviations and their corresponding full forms into a table, which is included in the Introduction section. Thank you for the suggestion.

7. Response to comment: What is the function of cathode luminescence in the pore structure classification?

Thank for you suggestion, We have made the necessary additions in our manuscript on line 115-120.

Reviewer #2: 

Thanks for your comments on our paper very much. It is a great encouragement to our research. The majority of manuscript is revised according by your complete comments and your comments bring us lot thinking on further research. The revisions are mainly focused on following aspects.

1. Response to comment: The abstract of the manuscript is very detailed. However, sometimes such abstract cannot clearly reflect the key insights and particularities of the research. Therefore, it is suggested that the authors optimize the structure of the abstract to make it more concise.

Response: 

We greatly appreciate the reviewer's valuable feedback regarding the abstract section of our manuscript. Taking their suggestion into consideration, we have carefully revised and condensed the abstract to ensure its conciseness and clarity. This modification will provide readers with a more succinct summary of our research findings. Thank you for the suggestion.

2. Response to comment: References should be supplemented with the latest research results. The introduction section lacks support of recent years. It is suggested that authors update references. Due to the lack of support from relevant references, it is impossible to determine whether the author has tracked the latest progress, which may lead to problems in summarizing the current situation and sorting out the problems. It is very important for readers to understand the research background.

Response: Thank for you suggestion, We have added the recent reference in our manuscript on line 50-51.

3. Response to comment: The text in Figure 1 should be in Times New Roman font. This Figure can be further optimized. The scale in Figure 3 cannot be seen clearly. Part of the text in Figure 6 is too small to read. Fig.8 is not clear, and the content cannot be seen clearly. Some text sizes in Figure 10 exceed the main text. They're lacks of explaining the Figure 11. please describe more.

Response: 

After checking all figures carefully，Our files have met the rules for PLOS ONE journal. Here below the rules. Thank you for your suggestion.

4. Response to comment: Tables 2 and 5 have inserted figures in the table. Perhaps it would be more appropriate to directly modify it to a highly summarized figure. The authors can consider whether modifications are needed (this is only a suggestion)

Response: Thank you for your suggestion.

5. Response to comment: Please further revise the statements in the paper that are difficult to read. Especially the naming of some new classifications. Part of the paper's discussion is not very user-friendly to readers and can be further optimized for expression. Commonly the abbreviations are explained the first time of the appearance of the corresponding terms. Here the abbreviations MICP, PCA, LDA, appearing in abstract, should be attributed to specific terms.

Response:

In accordance with the reviewer's suggestion, we have now included the full names of the abbreviations upon their first occurrence in the text and we have compiled all abbreviations and their corresponding full forms into a table, which is now included in the Introduction section. Thank you for the suggestion.

6.Response to comment: Please add more description of the new findings and significance of research. Especially in the abstract and conclusion section.

Response: we have added it in the abstract and conclusion.

---

## [Decision Letter · Decision Letter 1]

24 Apr 2024

PONE-D-23-28579R1Improved Pore Structure Characterization and Classification of Strong Diagenesis Sandstones by Data-mining Analytics in Tazhong Area, Tarim BasinPLOS ONE

Dear Dr. Wang,

Thank you for submitting your manuscript to PLOS ONE. After careful consideration, we feel that it has merit but does not fully meet PLOS ONE’s publication criteria as it currently stands. Therefore, we invite you to submit a revised version of the manuscript that addresses the points raised during the review process.

After another round of reviews, there are still some issues which must be addressed prior to the paper acceptance. In particular, the following issues need to be considered:

a) There is an apparent confusion about MICP results and their use in the paper. The authors indicate they were not suitable but apparently use it anyway. Please clarify;

b) The petrology characteristics of sandstone should be better described;

c) The results should be discussed in some detail;

d) There are problems with the formatting of the manuscript and with the English level. Please revise all of these before resubmitting.

We look forward to receiving your revised manuscript.

Kind regards,

Luan Carlos de Sena Monteiro Ozelim, D.Sc.

Academic Editor

PLOS ONE

Additional Editor Comments:

Dear authors, after another round of reviews, there are still some issues which must be addressed prior to the paper acceptance. In particular, the following issues need to be considered:

a) There is an apparent confusion about MICP results and their use in the paper. The authors indicate they were not suitable but apparently use it anyway. Please clarify;

b) The petrology characteristics of sandstone should be better described;

c) The results should be discussed in some detail;

d) There are problems with the formatting of the manuscript and with the English level. Please revise all of these before resubmitting.

Reviewers' comments:

Reviewer's Responses to Questions

**Comments to the Author**

1. If the authors have adequately addressed your comments raised in a previous round of review and you feel that this manuscript is now acceptable for publication, you may indicate that here to bypass the “Comments to the Author” section, enter your conflict of interest statement in the “Confidential to Editor” section, and submit your "Accept" recommendation.

Reviewer #3: All comments have been addressed

Reviewer #4: All comments have been addressed

2. Is the manuscript technically sound, and do the data support the conclusions?

Reviewer #3: Partly

Reviewer #4: Yes

3. Has the statistical analysis been performed appropriately and rigorously? 

Reviewer #3: Yes

Reviewer #4: No

4. Have the authors made all data underlying the findings in their manuscript fully available?

Reviewer #3: No

Reviewer #4: Yes

5. Is the manuscript presented in an intelligible fashion and written in standard English?

Reviewer #3: Yes

Reviewer #4: No

6. Review Comments to the Author

Reviewer #3: Dear Editor and authors

After Greetings,

Please find here in my comments on the manuscript number PONE-D-23-28579_R1.

This manuscript is well organized and easy to follow. It investigated an interesting research topic relevant to characterization and classification of pore structure in tight sandstone reservoirs. The topic fits well with the journal interest and it is of importance in tight sandstone reservoir. Authors attempted to provide pertinent details throughout the manuscript. However, I have some problems with the presentation and the arguments. The comments and suggestions are as follows:

1. Introduction: Author should introduce some new research trend on the characterization and classification of pore structure in tight sandstone reservoirs. Besides, it is necessary to clarify the importance of pore structure in tight sandstone.

2. Geological setting. Author should add some details and caption on each picture in the Fig.1 Where is the TZ11 and TZ12?

3. Geological setting. Line 65 what is “It” mean?

4. Geological setting. Line 65-68. The lithological characteristics of Jurassic system should be clarified carefully in the text.

5. Geological setting. Line 68-line 70 : Some references are needed to support the sedimentary facies.

4. Methodology

Author should explain the experimental analysis (e.g. MICP) or the source of experimental data.

5. Petrology characteristics of sandstone. This part is too simple and confusing. There is no caption and annotation in the Fig.5. The petrology characteristics of sandstone may be different in different Formation or member.

6. There is no discussion part in the manuscript. The structure of the manuscript need to be reorganized.

7. There are many language issues in the manuscript. The level of English of your manuscript does not meet the international journal's standard.

Reviewer #4: After carefully reading this article, there are some questions and suggestions regarding this article, as follows:

1. The use of “The Silurian system” ‘and “Sedimentation-diagenesis controls” as keywords in this article is inappropriate. It is recommended that the author readjust the keywords of this article;

2. In line 39, the author mentioned that analyzing MICP data does not reflect the real pore type, but the classification of pore structure in this article is based on MICP data. Does the MICP data analysis used in the article specifically adopt certain changes?

3. In lines 42-44, the shortcomings of nuclear magnetic technology and CT imaging techniques are mentioned, but the author's expression is not clear.

4. The list of nomenclature in line 61, since the full names and their corresponding abbreviations have already been mentioned earlier, there is no need to place them separately in the table;

5. The Pd, Smin, and Rd in the text should be written as Pd, Smin, and Rd, expressed using subscripts; Mpa should be written as MPa, and many physical quantities mentioned in the text have obvious errors in their units;

6. In lines 107-108, it is mentioned that " It can be classified into four rock types ", but why are there five types of rocks in Figures 4 (a) and (b)? What components do Q, F, and R represent in Figure 5? Suggest providing an explanation;

7. Placing XRD data in the reference list in lines 113-120 is deemed inappropriate, and it is recommended that the author revise it;

8. In Figure 8, the width of each column in the bar chart is different, and it is recommended to set it to a uniform width;

9. The use of references in this article is not standardized. The connected references in lines 41 and 42 should be placed in the same parentheses, and the same issue occurs in line 78; Lines 338-340 introduce the author's work and there is no need to add references;

10. The analysis and conclusion section of this article does not highlight the importance and innovation of the work done in the article;

11. The language of this article should be polished to improve its readability. And one more thing, is Wulumuqi in the address of the corresponding author the right English name of the city? And the second address is not complete.

7. PLOS authors have the option to publish the peer review history of their article (what does this mean?). If published, this will include your full peer review and any attached files.

Reviewer #3: No

Reviewer #4: No

---

## [Author Response · Author response to Decision Letter 1]

16 Jun 2024

Responds to the reviewer’s comments:

Reviewer #3: 

Thanks for your comments on our paper very much. It is a great encouragement to our research. The majority of manuscript is revised according by your complete comments and your comments bring us lot thinking on further research. The revisions are mainly focused on following aspects.

1. Response to comment: Introduction: Author should introduce some new research trend on the characterization and classification of pore structure in tight sandstone reservoirs. Besides, it is necessary to clarify the importance of pore structure in tight sandstone.

Response: 

We apologize for the oversight and have duly included the current research trend in the characterization and classification of pore structure in tight sandstone reservoirs in lines 33-36 as per your suggestion. Furthermore, we agree that it is essential to clarify the importance of pore structure in tight sandstone reservoirs, and have incorporated this in lines 31-33 as advised. Thank you for your valuable input.

2. Response to comment: Geological setting. Author should add some details and caption on each picture in the Fig.1 Where is the TZ11 and TZ12?.

Response: 

We strongly agree with the reviewer suggestion and we have added some details to Fig.1 to illustrate the location of TZ11 and TZ12 wellblock. Thank you for your suggestion.

3. Response to comment: Geological setting. Line 65 what is “It” mean?

Response: 

In this context, "it" refers to the stratigraphic formation, and we acknowledge that it was our oversight not to express this clearly. We have made the necessary adjustment by replacing "it" with "The stratigraphic formation" to improve clarity in the expression in line 74. Thank you for pointing this out.

4. Response to comment: Geological setting. Line 65-68. The lithological characteristics of Silurian system should be clarified carefully in the text.

Response: 

We greatly appreciate the reviewer's suggestion and have incorporated a detailed description of the lithological characteristics of the Silurian strata in lines 76-83. Thank you for the suggestions.

5. Response to comment: Geological setting. Line 68-line 70: Some references are needed to support the sedimentary facies.

Response:

We are referencing the sedimentary facies conclusions from the literature; However, due to our oversight, we neglected to cite the references. We have now added the citations in the text to support the sedimentary facies in lines 86-87.Thank you for the suggestion.

6. Response to comment: Methodology .Author should explain the experimental analysis (e.g. MICP) or the source of experimental data.

Response: 

We fully agree with the reviewer's feedback and have included a section explaining the analysis of the mercury injection capillary pressure (MICP) experiments, along with clarifying the source of the experimental data in lines 92-99.Thank you for the suggestion.

7. Response to comment: Petrology characteristics of sandstone. This part is too simple and confusing. There is no caption and annotation in the Fig.5. The petrology characteristics of sandstone may be different in different Formation or member.

Response: 

We apologize for not labeling clearly, the appropriate caption have been added. The explanation for Figure.5 is presented below in Lines 145-147. The title of Figure 5 has been expanded to include detailed annotations. The study zone for this research in this section focuses on the Upper-3rd submember of the upper Kepingtag member, so we don’t pay attention to the comparison with other submembers. It can be observed from Figure.5 that the petrology characteristics of the sandstone are consistent within this submember. Thank you for the suggestion.

8. Response to comment: There is no discussion part in the manuscript. The structure of the manuscript needs to be reorganized.

Response: 

It was an oversight on our part not to label which section is the discussion. In this paper, we have integrated the discussion and results. For instance, in section 4.3, we discuss the relative independence of the information about the impact of slope in the scree plot on various features. Similarly, in section 4.4, we discuss the distribution trends of Pd pore-throat pressures and the selectivity trend distributions, among others. We have renamed the entire fourth section as "Results and Discussion." Thank you for the suggestion.

9. Response to comment: There are many language issues in the manuscript. The level of English of your manuscript does not meet the international journal's standard.

Response: 

We have conducted a thorough review of the entire manuscript and addressed numerous language issues, such as those in lines 37-38,39-44,63-64,65-68,104-105,116-117,216-217 307-308 ,237-238 and 412-414 among others (marked in the text; not listed here). This has enhanced the English proficiency of the manuscript. We appreciate your feedback, which brought this to our attention, and we have made the necessary revisions based on your suggestions.

Reviewer #4: 

Thanks for your comments on our paper very much. It is a great encouragement to our research. The majority of manuscript is revised according by your complete comments and your comments bring us lot thinking on further research. The revisions are mainly focused on following aspects.

1. Response to comment: The use of “The Silurian system”and “Sedimentation-diagenesis controls” as keywords in this article is inappropriate. It is recommended that the author readjust the keywords of this article.

Response: 

We appreciate your suggestion, and upon careful consideration, we agree that using "Silurian system" and "sedimentary-diagenetic control" as keywords may not be appropriate. Accordingly, we have replaced these terms with "strong diagenesis sandstone" and "low-permeability reservoir" as per your recommendation. Thank you for your valuable input.

2. Response to comment: In line 39, the author mentioned that analyzing MICP data does not reflect the real pore type, but the classification of pore structure in this article is based on MICP data. Does the MICP data analysis used in the article specifically adopt certain changes?

Response: 

We feel sorry for the previous words to make reviewers misunderstand. The essence of this article is to highlight that simple statistical description and regression analysis based on mercury injection capillary pressure (MICP) data often fall short in capturing the true pore types. This limitation stems from the inadequacy of overly simplistic models in effectively classifying pore structures and the significant impact of subjective human factors on the low efficiency of simple statistical descriptions, leading to inaccurate pore type characterization. In this study, we have employed an advanced approach with machine learning methods and models based on MICP data to achieve the classification and characterization of pore structures. This approach allows for a more comprehensive and accurate processing of data, minimizing the influence of subjective factors and enabling detailed analysis and classification of the data, thus facilitating precise classification and characterization of pore structures. Thank you for bringing this to our attention.

3. Response to comment: In lines 42-44, the shortcomings of nuclear magnetic technology and CT imaging techniques are mentioned, but the author's expression is not clear.

Response:

We have revised the discussion on the limitations of nuclear magnetic technology and CT imaging techniques in lines 48-52. Thank you for the suggestion.

4. Response to comment: The lists of nomenclature in line 61, since the full names and their corresponding abbreviations have already been mentioned earlier, there is no need to place them separately in the table.

Response:

Based on your suggestion, we acknowledge that the nomenclature table was indeed redundant, and we have removed it. Thank you for the suggestion.

5. Response to comment: The Pd, Smin, and Rd in the text should be written as Pd, Smin, and Rd, expressed using subscripts; Mpa should be written as MPa, and many physical quantities mentioned in the text have obvious errors in their units.

Response:

Thank you for bringing this to our attention. We have rectified the formatting issue, and now it is correctly written as Pd,Smin,Rd and MPa. Additionally, we have corrected the erroneous units of the physical quantities mentioned in the article. Thank you for the suggestion.

6. Response to comment: In lines 107-108, it is mentioned that " It can be classified into four rock types ", but why are there five types of rocks in Figures 4 (a) and (b)? What components do Q, F, and R represent in Figure 5? Suggest providing an explanation.

Response:

There are actually five rock types. It was our mistake to list them as four, and we have now made the necessary correction in line 131. Thank you for pointing that out. Q represents Quartz, F represents Feldspar, and R represents Rock Fragments. The QFR ternary plot is a graphical method used to describe the proportions of clastic particles in sedimentary rocks. In academic papers on sandstone reservoir studies, the sandstone classification ternary plot is widely used to differentiate sandstone types. We have now added the full name "QFR ternary plot" to Figure 5(a) and Figure 5(b) .Thank you for the suggestion.

7. Response to comment: Placing XRD data in the reference list in lines 113-120 is deemed inappropriate, and it is recommended that the author revise it.

Response:

It's inappropriate to include XRD data in the reference list as its conclusions are derived from previous studies. Therefore, we have removed the XRD data from this section and directly referenced the previous studies in line 138.Thank you for the suggestion.

8. Response to comment: In Figure 8, the width of each column in the bar chart is different, and it is recommended to set it to a uniform width.

Response:

There may be some confusion here, Fig. 8 is not a bar chart, and it is a cluster dendrogram that has already been standardized. Thank you very much for your suggestion.

9. Response to comment: The use of references in this article is not standardized. The connected references in lines 41 and 42 should be placed in the same parentheses, and the same issue occurs in line 78; Lines 338-340 introduce the author's work and there is no need to add references.

Response:

Thank you for pointing out the issue with the referencing format. We have now placed the linked references in the brackets in line 48, and the same issue has been rectified in Line 103 as well. We fully agree that the introduction of the author's work does not require the addition of references, and it has already been deleted. Thank you very much for your suggestion.

10. Response to comment: The analysis and conclusion section of this article does not highlight the importance and innovation of the work done in the article.

Response:

We fully agree with your opinion and indeed felt that the conclusion lacked some explanation regarding the importance and innovation of our work in Line 451-455. Following your advice, we have added emphasis on the importance and innovation in the conclusion. Thank you very much for your suggestion, which has improved the article.

11. Response to comment: The language of this article should be polished to improve its readability. And one more thing, is Wulumuqi in the address of the corresponding author the right English name of the city? And the second address is not complete.

Response:

Thank you for bringing this to our attention. We have thoroughly reviewed the language structure of the entire article and addressed numerous language-related issues, thus enhancing the English proficiency and readability of the manuscript, as exemplified in lines 37-38 and 63-63. We have conducted a thorough review of the entire manuscript and addressed numerous language issues, such as those in lines 37-38,39-44,63-64,65-68,104-105,116-117,216-217 307-308 ,237-238 and 412-414 among others (marked in the text; not listed here). Furthermore, Wulumuqi is a correct English name for the city, and it is more often written as Urumqi. Following your suggestion, we have completed the second address accordingly. Thank you very much for your suggestion. 

We appreciate for Editors/Reviewers’ warm work earnestly, and hope that the correction will meet with approval.

Once again, thank you very much for your comments and suggestions.

---

## [Decision Letter · Decision Letter 2]

6 Aug 2024

Improved Pore Structure Characterization and Classification of Strong Diagenesis Sandstones by Data-mining Analytics in Tazhong Area, Tarim Basin

PONE-D-23-28579R2

Dear Dr. Wang,

We’re pleased to inform you that your manuscript has been judged scientifically suitable for publication and will be formally accepted for publication once it meets all outstanding technical requirements.

Kind regards,

Luan Carlos de Sena Monteiro Ozelim, D.Sc.

Academic Editor

PLOS ONE

Additional Editor Comments (optional):

Reviewers' comments:

Reviewer's Responses to Questions

**Comments to the Author**

1. If the authors have adequately addressed your comments raised in a previous round of review and you feel that this manuscript is now acceptable for publication, you may indicate that here to bypass the “Comments to the Author” section, enter your conflict of interest statement in the “Confidential to Editor” section, and submit your "Accept" recommendation.

Reviewer #3: All comments have been addressed

Reviewer #4: All comments have been addressed

2. Is the manuscript technically sound, and do the data support the conclusions?

Reviewer #3: Yes

Reviewer #4: Yes

3. Has the statistical analysis been performed appropriately and rigorously? 

Reviewer #3: Yes

Reviewer #4: Yes

4. Have the authors made all data underlying the findings in their manuscript fully available?

Reviewer #3: Yes

Reviewer #4: Yes

5. Is the manuscript presented in an intelligible fashion and written in standard English?

Reviewer #3: Yes

Reviewer #4: Yes

6. Review Comments to the Author

Reviewer #3: This manuscript can be accepted in present state. However, there is no caption and annotation in the Fig.5

Reviewer #4: After reading the author's revised manuscript, I have noticed that the author has seriously considered and made corresponding revisions to the issues and suggestions raised before. These revisions have improved the quality of the paper to some extent. Based on the current content and quality, I believe that this paper has met the basic requirements for publication. Therefore, I agree with the publication of this paper.

7. PLOS authors have the option to publish the peer review history of their article (what does this mean?). If published, this will include your full peer review and any attached files.

Reviewer #3: No

Reviewer #4: No

---

## [Editor Report · Acceptance letter]

16 Aug 2024

PONE-D-23-28579R2 

PLOS ONE

Dear Dr. Wang, 

I'm pleased to inform you that your manuscript has been deemed suitable for publication in PLOS ONE. Congratulations! Your manuscript is now being handed over to our production team.

Kind regards, 

on behalf of

Dr. Luan Carlos de Sena Monteiro Ozelim 

Academic Editor

PLOS ONE